# AI-guided Cas9 engineering provides an effective strategy to enhance base editing

Dongyi Wei[1,8], Peng Cheng [2,8], Ziguo Song[3,8], Yixin Liu[1], Xiaoran Xu [1], Xingxu Huang [4,5], Xiaolong Wang[3,6], Yu Zhang [7✉], Wenjie Shu [2✉] & Yongchang Wei [1✉]

## Abstract

**Precise genome editing is crucial for functional studies and therapies. Base editors, while powerful, require optimization for efficiency. Meanwhile, emerging protein design methods and protein language models have driven efficient and intelligent protein engineering. In this study, we employed the Protein Mutational Effect Predictor (ProMEP) to predict the effects of single-site saturated mutations in Cas9 protein, using AncBE4max as the prototype to construct and test 18 candidate point mutations. Based on this, we further predicted combinations of multiple mutations and successfully developed a high-performance variant AncBE4max-AI-8.3, achieving a 2–3-fold increase in average editing efficiency. Introducing the engineered Cas9 into CGBE, YEE-BE4max, ABE-max, and ABE-8e improved their editing performance. The same strategy also substantially improves the efficiencies of HF-BEs. Stable enhancement in editing efficiency was also observed across seven cancer cell lines and human embryonic stem cells. In conclusion, we validated that AI models can serve as more effective protein engineering tools, providing a universal improvement strategy for a series of gene editing tools.**

**Keywords** Artificial Intelligence; Base Editor; Gene Editing; hESCs; Protein Engineering
**Subject Category** Biotechnology & Synthetic Biology

## Introduction

The emergence of clustered regularly interspaced short palindromic repeats (CRISPR)/Cas system has revolutionized genome editing technology. In the decade since CRISPR-Cas9 was confirmed by numerous researchers as a highly efficient mammalian genome editing tool, many Cas9-based derivative systems have been developed, and

more types of Cas proteins have been discovered, making it possible to edit specific base pairs or DNA fragments in a wider range of situations (Anzalone et al, 2019; Gilbert et al, 2014; Komor et al, 2016; Makarova et al, 2011; Sander and Joung, 2014; Wang and Doudna, 2023). Typically, Cas9-induced DSBs are used for gene knockout rather than precise correction of point mutations (Mali et al, 2013). Base editors (BEs), a new generation of gene editing tools based on nickase or dead Cas protein fused with a cytidine or adenosine deaminase, have emerged to address this limitation (Komor et al, 2016; Rees and Liu, 2018). The editing process of BEs is similar to that of Cas9 except for the target base deamination in the R-loop and the absence of DSB (Anzalone et al, 2020). Typical base editors include cytosine base editors (CBEs) and adenine base editors (ABEs) (Gaudelli et al, 2017; Komor et al, 2016), which enable C:G to T:A or A:T to G:C conversions and could potentially correct ~30% of currently annotated human pathogenic variants (Landrum et al, 2016). While the advent of prime editing (PE) could overcome this limitation, its relatively lower editing efficiency compared to BE remains an obstacle to broader application (Anzalone et al, 2019). Numerous base editors based on glycosylase have been reported to expand editing types. For example, CGBE, gTBE, and AYBE enable conversions such as C:G to G:C, T:A to G:C or A:T to T:A editing, respectively, filling the gaps in mutation correction capabilities that were not covered by traditional base editors (Chen et al, 2024; Koblan et al, 2021; Kurt et al, 2020; Tong et al, 2023). Regardless of the catalytic function provided by deaminases or glycosylases in base editing, Cas9 remains the fundamental backbone of the base editing tools.

Despite its proven potential, efficiency is one of the basic obstacles hindering the use of BEs for molecular therapy and in wider biotechnological applications. To enhance editing efficiency, previous studies have primarily focused on optimizing deaminases or other elements within the base editing systems. Common strategies include the engineering of specific deaminases to improve their catalytic activity (Chen et al, 2022; Gaudelli et al, 2020; Richter et al, 2020; Xie et al, 2024), refining the linker region between the Cas9 protein and the deaminase, incorporating single-stranded DNA-binding proteins to stabilize the deaminase's binding to ssDNA, and modifying nuclear localization signals (Kleinstiver

[1]Department of Radiation and Medical Oncology, Zhongnan Hospital of Wuhan University, Wuhan University, Wuhan, China. [2]Bioinformatics Center of AMMS, Beijing, China. [3]International Joint Agriculture Research Center for Animal Bio-Breeding of Ministry of Agriculture and Rural Affairs, College of Animal Science and Technology, Northwest A&F University, Yangling 712100 Shaanxi, China. [4]School of Life Science and Technology, ShanghaiTech University, Shanghai 201210, China. [5]Zhejiang Provincial Key Laboratory of Pancreatic Disease, The First Affiliated Hospital, and Institute of Translational Medicine, Zhejiang University School of Medicine, Hangzhou, Zhejiang, China. [6]Key Laboratory of Livestock Biology, Northwest A&F University, Yangling 712100 Shaanxi, China. [7]Shanghai-MOST Key Laboratory of Health and Disease Genomics,NHC Key Lab of Reproduction Regulation, Shanghai Institute for Biomedical and Pharmaceutical Technologies, Shanghai 200237, China. [8]These authors contributed equally: Dongyi Wei, Peng Cheng, Ziguo Song. ✉E-mail: zhangy@sibpt.cn; shuwj@bmi.ac.cn; weiyongchang@whu.edu.cn

et al, 2016; Koblan et al, 2018; Slaymaker et al, 2016; Wang et al, 2017). In parallel, numerous studies have also focused on optimizing the Cas9 protein in base editing systems, but the purpose has shifted from efficiency improvement to optimizing genome coverage or minimizing off-target activity (Chen et al, 2017; Hu et al, 2018; Kleinstiver et al, 2015; Walton et al, 2020). For instance, the development of high-fidelity Cas9 variants, such as HF1, has been a major breakthrough in addressing concerns about off-target effects (Kleinstiver et al, 2016). HF1 incorporates a series of rationally designed mutations that reduce non-specific DNA binding, significantly improving the safety profile of the editor. However, this improvement often comes at the cost of reduced on-target editing efficiency, which limits the broader application of high-fidelity editors in therapeutic contexts. This efficacy-safety trade-off remains a critical barrier for safe and effective in vivo genome editing. Compared with previous efforts that apply a BE-specific strategy for efficiency improvement, we hypothesized that a high-performance Cas9 variant could provide a one-size-fits-all solution for a series of BEs. However, designing such a universal Cas9 is extremely challenging regarding the multi-domain structure, gigantic mutation space, and complex mutation epistasis (Miton and Tokuriki, 2016; Nishimasu et al, 2014; Olson et al, 2014). Traditional directed evolution methods and rational design strategies demonstrate notable achievements in protein engineering, but these processes remain inefficient and labor-intensive (Packer and Liu, 2015; Slaymaker et al, 2016).

Driven by the advancements in large-scale data, computing power and algorithms, de-novo protein design methods and protein language models emerge as a fast and intelligent approach to design proteins with enhanced functions (Biswas et al, 2021; Dauparas et al, 2022; Madani et al, 2023; Ruffolo and Madani, 2024; Yeh et al, 2023). We recently developed a protein mutational effect predictor (ProMEP) (Cheng et al, 2024), a zero-shot mutation effects prediction method that simultaneously learns sequence- and structure-context from 160 million proteins. By navigating the protein fitness landscape that describes the sequence-function mapping (Chen et al, 2023; Ding et al, 2019), ProMEP identifies fitness peaks and guides protein engineering. Nevertheless, the design of a universal and high-performance Cas9 protein remains uncertain.

Here we used ProMEP to design high-performance Cas9 mutants optimized for base editing. Using AncBE4max editor as a prototype, we subsequently applied the most efficient AI-designed Cas9 mutant to various CBEs and ABEs, and validated its editing performance and specificity across multiple cancer cell lines and human embryonic stem cells (hESCs). Overall, this research introduces an AI-generated Cas9 mutant as a generalizable strategy to enhance the editing efficiency of diverse base editing tools.

## Results

### AI-guided engineering of the Cas9 protein

In our prior study, we developed ProMEP (Cheng et al, 2024), a multimodal method that enables zero-shot prediction of mutation effects. In contrast to protein language models that solely utilize sequence data (Ferruz et al, 2022; Meier et al, 2021; Notin et al, 2022), ProMEP employs a multimodal architecture integrating structural information critical for protein function. It has demonstrated great capability in guiding protein engineering by depicting protein fitness landscapes and accurately identifying protein variants with high fitness scores. Specifically, ProMEP uses the sequence and structure of a wild-type protein as input and employs a customized protein point cloud to extract protein structural information at atomic resolution. It then applies a rotation- and translation-equivariant structure embedding module to simulate interactions among spatially adjacent amino acids. The output of ProMEP can be represented as an L × 20 matrix that depicts the probability distribution of 20 amino acids at various positions within a protein. Consequently, the fitness score of a protein variant can be interpreted as the probability difference between the mutated sequence and the wild-type sequence. Protein variants with high fitness scores can be chosen to guide protein engineering.

To design a high-performance Cas9 variant (Fig. 1A), we first constructed a virtual single-point saturation mutagenesis library containing 25,992 single mutants. Then, we used ProMEP to calculate the fitness scores for all these mutants and ranked them accordingly (Fig. EV1A and Table EV1). Enrichment analysis showed that X-to-K mutants were significantly enriched in the top 5% of mutants ($p$ value <0.0001; two-sided $T$-test) (Fig. EV1B). In addition, we detailed the predicted probability distributions of all amino acid types at altered positions within Cas9 (Fig. EV2). Based on the enrichment analysis, we selected 18 mutants via a hybrid approach combining fitness score thresholds with mutation-type quotas for experimental validation (Methods).

We introduced the corresponding amino acid mutations into the nCas9 based on the AncBE4max editor, constructing 18 single-mutant variants (Cas9-AI-1). Subsequently, we co-transfected these variants along with corresponding sgRNA plasmids targeting numerous endogenous loci into HEK293T cells, using mCherry fluorescence as a marker for successful transfection. Approximately 48 h post-transfection, we enriched the mCherry-positive cells (top 15%) via flow cytometry. After extracting genomic DNA, we subjected them to next-generation sequencing (NGS). NGS results indicated that ProMEP demonstrates high confidence in predicting top-ranked beneficial mutations. Some single mutants (e.g., G1218R, G1218K, and C80K) performed well, with editing efficiencies higher than the wild-type version across all tested endogenous sites (Fig. 1B) (targets not presented in the multi-C target are displayed in Appendix Fig. S1a). For instance, the G1218R variant showed a 1.64-fold average editing activity than the wild-type, with its maximum activity reaching 2.33-fold. Meanwhile, some single mutants (e.g., T622W, V1015K, C80M, C80R, and C574K), although slightly lower than the wild-type at certain sites, outperformed the wild-type at most sites (Fig. 1C). Indel ratios were also analyzed, revealing consistently low overall incidence rates across genomic sites, with an average of less than 1% (Appendix Fig. S1b). This demonstrates that fitness score analysis could effectively guide the selection of positive mutations, significantly narrowing the scope for experimental validation.

### AI-generated multi-mutant Cas9 protein with enhanced base editing performance

Typically, combining positive mutations yields improved editing efficiency. Eight positive mutations, including G1218R, T622W, V1015K, G1218K, C80K, C80M, C80R, and C574K, exhibited higher average efficiency than the wild-type (Fig. 1C). After eliminating redundant mutation sites (e.g., for position 1218, where

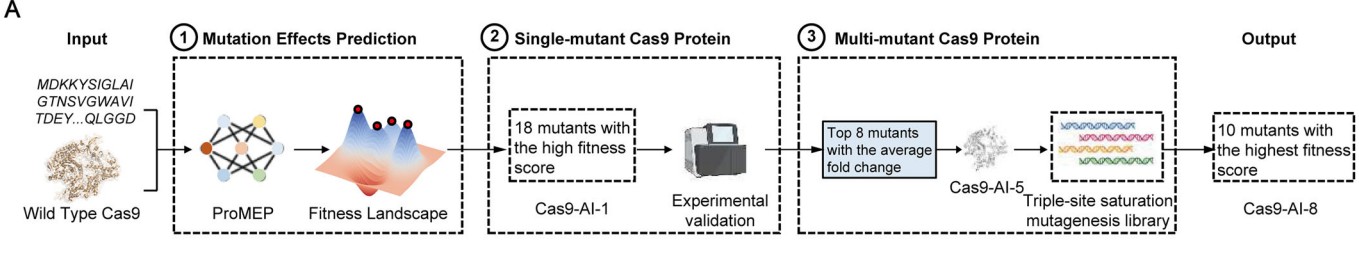

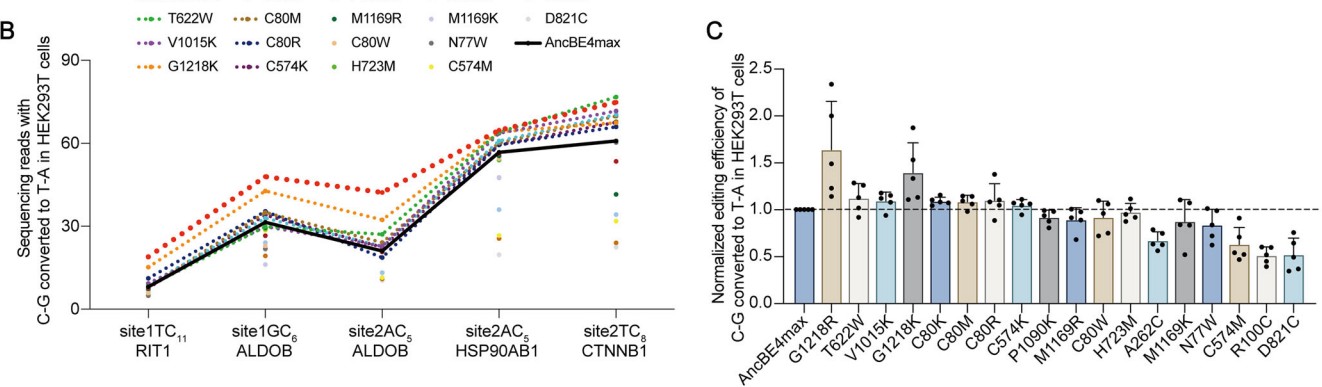

**Figure 1. AI-guided engineering of Cas9 proteins.**

(A) The screening workflow for AI-guided protein engineering. ProMEP utilizes both sequence and structural information of the wild-type Cas9 as input and depicts its fitness landscape to guide protein engineering. (B) Editing efficiencies of AncBE4max and 18 single mutants at the primary editing positions across five endogenous sites in HEK293T cells. The AncBE4max baseline is shown in black, and single mutants with higher editing efficiencies are highlighted in bold ($n = 3$ biological replicates). (C) Normalized editing efficiencies for all sites shown in (C) ($n = 5$). The editing level of AncBE4max at each site were set as 1 (indicated by horizontal dashed line). Data information: In (B, C), data were presented as mean ± s.d. Source data are available online for this figure.

both G1218K and G1218R mutations were present, we retained the mutation with higher editing efficiency), we selected G1218R, T622W, V1015K, C80K, and C574K to create a new variant (Cas9-AI-5) (Fig. 1A). To more thoroughly explore combinations of mutations that could improve base editing efficiency in Cas9, we used ProMEP to simulate the pairing of existing beneficial mutations with saturation mutations across all other positions within the protein. We constructed a three-site saturation mutagenesis library of $\sim 2.8 \times 10^{12}$ variants based on Cas9-AI-5, and used ProMEP to calculate the fitness scores for each multi-point mutant (Fig. EV1A). We selected the top ten variants with the highest fitness scores for further evaluation (Fig. 1A and Table EV1), and introduced these 10 eight-point combination mutations into AncBE4max, naming them AncBE4max-AI-8.n(1–10) (Appendix Table S1). To investigate whether the eight-point combination mutations exhibited higher editing efficiency, we co-transfected the 10 eight-point mutants along with the G1218R variant and the wild type in HEK293T cells, targeting four endogenous gene sites. Cells were harvested 48 h post-transfection, and the editing efficiency was analyzed by NGS (Fig. 2A and Appendix Fig. S2a). We further verified that G1218R showed superior performance compared to the wild type at all sites (shown with a yellow solid line), while most eight-point combination mutants demonstrated higher editing efficiency than the G1218R variant. Among them, (AncBE4max-AI-) 8.3, 8.4, 8.6, 8.8, and 8.10 achieved approximately twofold higher average efficiencies

compared to the wild type. AncBE4max-AI-8.3 exhibited the best performance within the group, averaging 2.34-fold higher than the wildtype, with no significant increase in indel rates ($p = 0.3339$, two-tailed, unpaired $T$-test) (Fig. 2B, and Appendix Fig. S2b).

To further validate the editing performance of AncBE4max-AI-8.3, we conducted additional tests at more endogenous sites in both HEK293T and HeLa cell lines (Fig. 2C,D) (all values for multi-C sites are presented in Appendix Figs. S3a, S4a). We observed that AncBE4max-AI-8.3 significantly enhanced editing efficiency in both cell lines, with average increases of 2.1-fold in HEK293T and 2.6-fold in HeLa cells, particularly at sites where the wild-type editing efficiency was suboptimal (the highest enhancement reached 3.0-fold (in HEK293T) and 5.0-fold (in HeLa)) (Fig. 2E). In terms of indel ratios, no significant differences were observed among these variants ($p > 0.05$, two-tailed, unpaired $t$-test) (Appendix Figs. S3b, S4b). Further analysis of the editing window revealed that AncBE4max-AI-8.3 only improved the editing efficiency within the original window, and did not change the range of the editing window in both cell lines (Fig. 2F; Appendix Fig. S4c). Therefore, we identified AncBE4max-AI-8.3 as the final version. The mutations in the nCas9 of AncBE4max-AI-8.3 are C80K; G1218R; T622W; C574K; V1015K; N501Y; M939F; and H723I (Appendix Table S1). Based on the resolved protein-sgRNA-DNA structure, it could be found that mutations in AncBE4max-AI-8.3 exhibited a relatively separate spatial distribution throughout the five protein domains (Appendix Fig. S5). In previous studies, Cas9

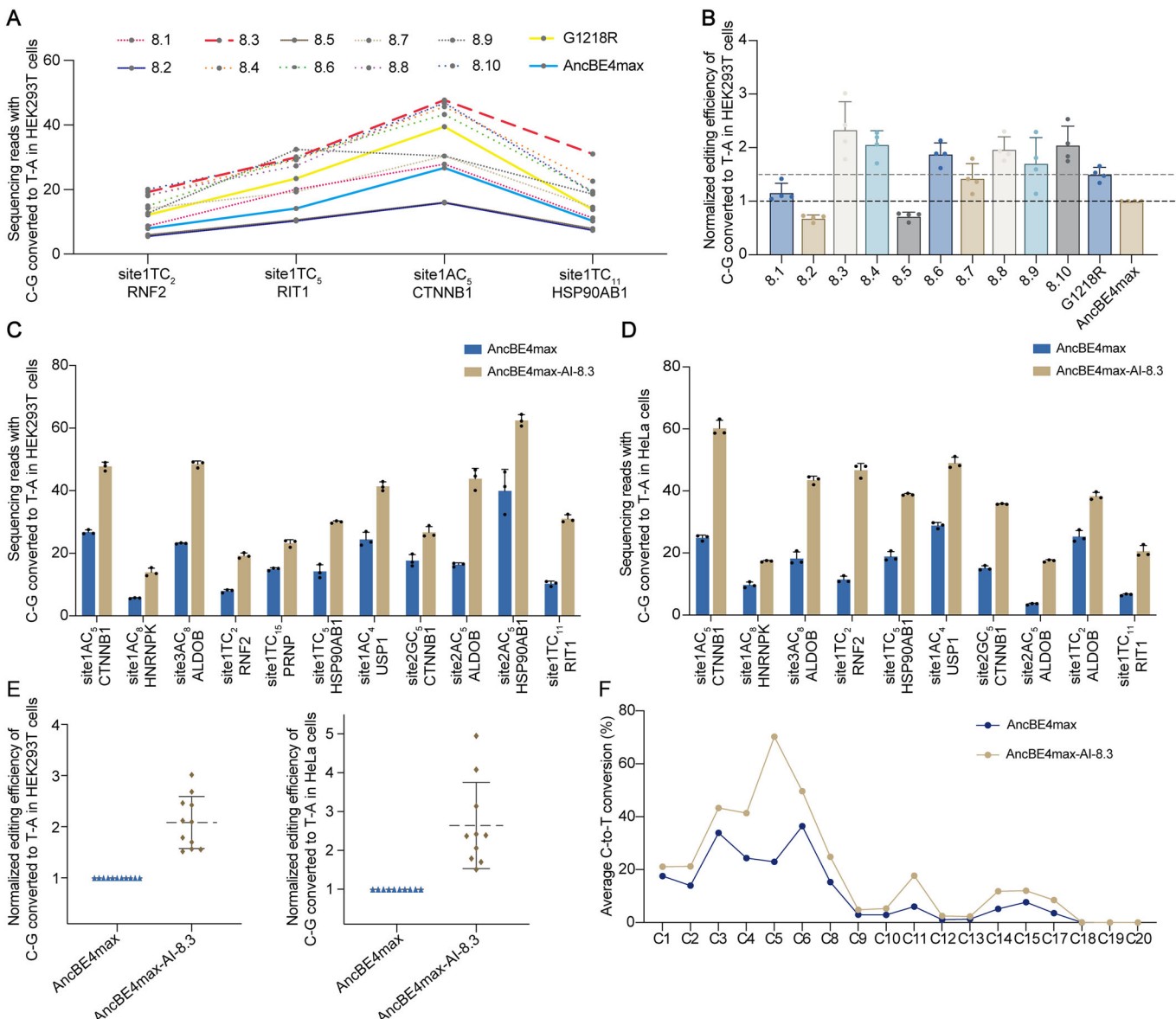

**Figure 2. Screening of AI-guided multi-mutant Cas9 protein.**

(A) Parallel comparison of editing efficiencies at primary editing positions between ten 8-point mutant variants, G1218R, and AncBE4max across four endogenous sites in HEK293T cells. Yellow and blue solid lines represent baseline levels of G1218R and AncBE4max. The red dashed line highlights AncBE4max-AI-8.3, which showed the highest average editing efficiency ($n = 3$ biological replicates). (B) Normalized editing efficiencies for all sites shown in (A) ($n = 4$). The editing level of AncBE4max at each site were set as 1 (positions of AncBE4max and G1218R are shown by black and brown dashed lines). (C) Editing efficiencies of AI-8.3 mutant and AncBE4max at primary editing positions across 11 endogenous loci in HEK293T cells ($n = 3$ biological replicates). (D) Similar experiments to (C) at ten genomic sites in HeLa cells ($n = 3$ biological replicates). (E) Normalized editing efficiencies for all sites shown in (C) ($n = 11$) and (D) ($n = 10$). (F) Summary of average C-to-T conversion (%) for AncBE4max and AncBE4max-AI-8.3 at protospacer positions (1–20) in 12 endogenous sites of HEK293T cells ($n = 12$). Data information: In (A–E), data were presented as mean ± s.d. Source data are available online for this figure.

engineering typically focuses on one or several key functional domains (Jiang et al, 2016; Slaymaker et al, 2016). The application of AI models has transcended these limitations, enabling a more holistic engineering approach. This allows us to minimize the possibility of missing crucial mutations or neglecting synergistic effects between domains. Overall, our results validate the model's predictive accuracy and successfully identify high-performance combinatorial mutations in the Cas9 protein for base editing applications.

## High-performance Cas9 variant as a universal strategy to enhance multiple versions of base editors

Since the Cas9 protein is widely used across various base editing tools for its role in DNA targeting and ssDNA substrate exposure, we hypothesized that our optimized Cas9 variants could provide a versatile engineering strategy applicable to multiple base editors. To elucidate this, we incorporated the high-performance Cas9 variant

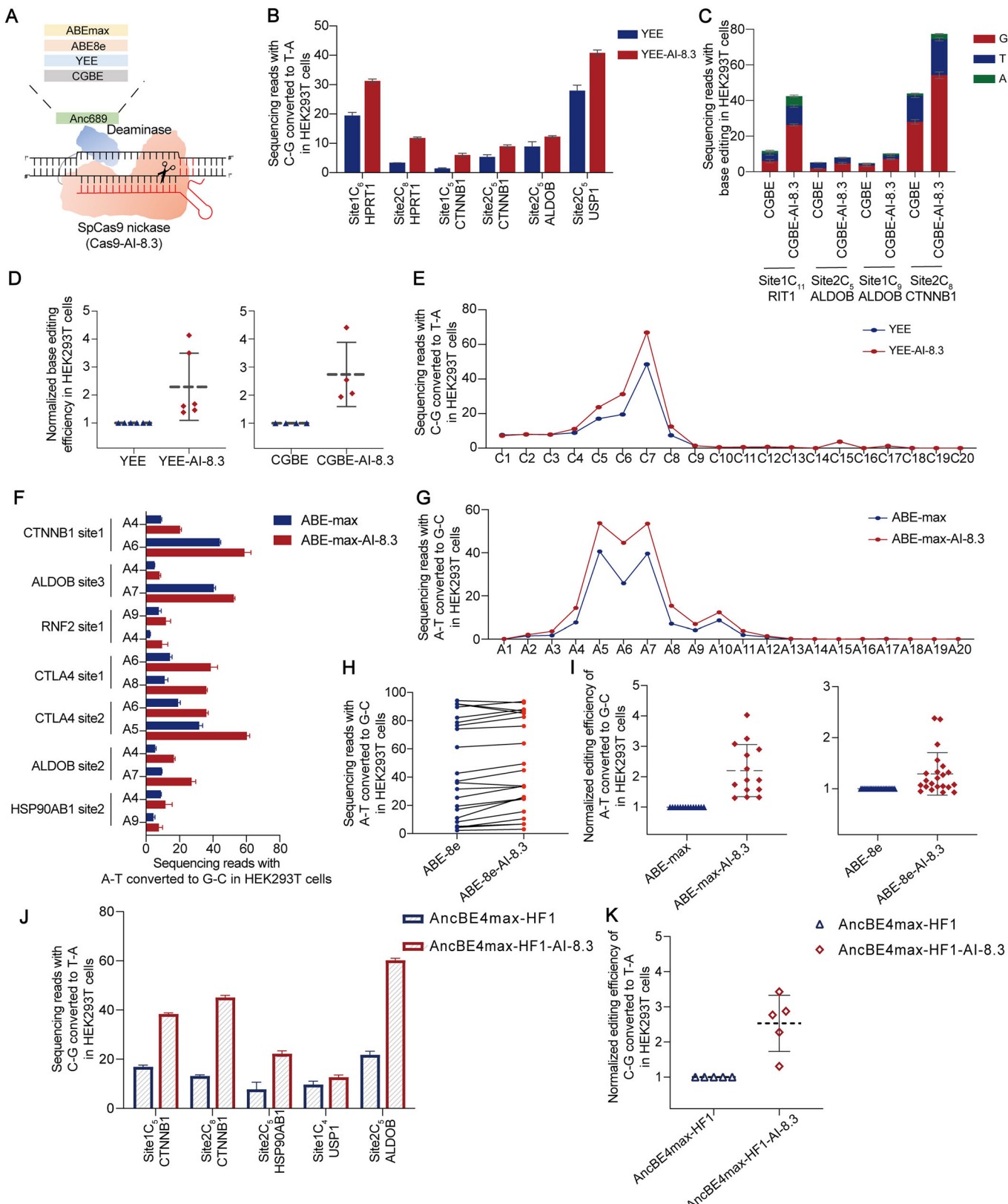

**Figure 3. Multiple versions of base editors enhanced by a high-performance Cas9 variant.**

(A) Schematic diagram of AI-engineered Cas9 variants combined with other deaminases. (B) Editing efficiencies of YEE-AI-8.3 and YEE-BE4max at primary editing positions across six endogenous loci in HEK293T cells ($n = 3$ biological replicates). (C) Total sequencing reads with the target C, respectively converted to T/G/A bases, are shown ($n = 3$ biological replications). (D) Normalized C-to-T conversion (%) for all sites shown in (B) ($n = 6$) and (C) ($n = 4$). (E) Summary of average C-to-T conversion (%) for YEE-BE4max and YEE-AI-8.3 at protospacer positions (1–20) in 13 endogenous sites of HEK293T cells ($n = 13$). (F) Sequencing reads with A-to-G of ABE-max and ABE-max-AI-8.3 at all editing positions across seven sites in HEK293T cells ($n = 3$ biological replicates). (G) Summary of average A-to-G conversion (%) for ABE-max and ABE-max-AI-8.3 at protospacer positions (1–20) in 15 endogenous sites of HEK293T cells ($n = 15$). (H) A per-site, paired comparisons between the editing efficiencies of ABE8e and ABE8e-AI-8.3 ($n = 22$ edited points). (I) Normalized A-to-G conversion (%) for all sites shown in (F) ($n = 14$) and (C) ($n = 22$). (J) Editing efficiencies of AncBE4max-HF1 and AncBE4max-HF1-AI-8.3 at primary editing positions across five sites in HEK293T cells ($n = 3$ biological replicates). (K) Normalized C-to-T conversion (%) for all sites shown in (J) ($n = 6$). Data information: In (B–D, F, I–K), data were presented as mean ± s.d. Source data are available online for this figure.

from AncBE4max-AI-8.3 into different versions of cytosine base editors, including YEE-BE4max, a high-fidelity CBE variant, and CGBE, which enables C-to-G base transversions (Fig. 3A) (Doman et al, 2020; Koblan et al, 2021). We renamed the engineered variants as YEE-BE4max-AI-8.3 (simplified as YEE-AI-8.3 in subsequent descriptions) and CGBE-AI-8.3. For YEE-AI-8.3, we targeted six genomic loci in HEK293T cells and observed enhanced editing efficiency at every C target site across all loci (average 2.3-fold), particularly at sites with initially low efficiency, such as HPRT1 site2 and CTNNB1 site1 (Fig. 3B,D; Appendix Fig. S6a). To assess whether the high-activity YEE-AI-8.3 altered the editing window, we analyzed the average C-to-T conversion (%) at each C target position within the protospacer sequences from 13 sites (Fig. 3E). The results indicated that the original editing window was not broadened, and our high-activity variant only exhibited improved editing activity at positions C4-C8. For CGBE-AI-8.3, we selected four genomic loci in HEK293T cells to evaluate overall base editing levels (C-to-G/T/A) (Fig. 3C; Appendix Fig. S7a). Compared to the wildtype, CGBE-AI-8.3 exhibited a significant improvement in C-to-G editing efficiency across all target sites, achieving an average increase of 2.7-fold (Fig. 3D). Meanwhile, we observed corresponding increases in C-to-T/A levels, though these increases were substantially lower than the C-to-G levels, and CGBE-AI-8.3 showed higher product purity (C-to-G) compared to WT (Appendix Fig. S7b).

Following successful validation in cytosine base editors, we extended our strategy to adenine base editors (ABE) by incorporating the engineered Cas9 variants into the most used versions: ABE-max and ABE8e (Koblan et al, 2018; Richter et al, 2020). For ABE-max, seven target sites were chosen to compare the performance between wildtype and ABE-max-AI-8.3 in HEK293T cells. NGS results indicated improved editing efficiency across all target sites, with an average increase of 2.2-fold over the wild-type. At sites with initially low editing efficiency, improvements reached up to 3-4-fold (Fig. 3F,I). Analysis of the editing window showed trends consistent with our previous findings (Fig. 3G), where the high-performance Cas9 variant enhanced editing efficiency within the original window without expanding it. For ABE8e, the highest-efficiency version, ABE8e-AI-8.3 also demonstrated improvement at most target sites. Specifically, the enhancement effects of ABE8e-AI-8.3 were mainly located at target sites with initial efficiencies below 80%, while sites with efficiencies above 85% reached an efficiency plateau for both ABE8e and ABE8e-AI-8.3 (Fig. 3H,I). Furthermore, the indel rates for all variants mentioned above are maintained at acceptably low levels (Appendix Figs. S6b, S7C, S8) ($p > 0.05$, two-tailed, unpaired T-test).

High base editing activity is often accompanied by off-target effects, which is a key safety challenge for current highly active base editors (BEs). Introducing high-fidelity nCas9 variants (such as HF1-nCas9) can effectively reduce Cas9-dependent off-target effects, but these variants are often limited in application due to insufficient activity (Kleinstiver et al, 2016). Therefore, we attempted to introduce our efficient mutation combinations into the HF1-nCas9-based BE system. We expected this approach to effectively address the issue of low activity while maintaining the desired low level of off-target effects. Our tests at five endogenous loci in HEK293T cells showed that the application of AncBE4max-HF1 indeed resulted in lower base editing activity (Figs. 2C, 3J and EV3A). Meanwhile, the introduction of the eight-mutation combination significantly alleviated its low on-target activity, achieving an average editing efficiency 2.5 times higher than the original HF1 version. (Fig. 3K) Subsequently, we predicted the potential off-target sites of three genomic loci, CTNNB1, HSP90AB1, and USP1, using the Cas-OFFinder website (http://www.rgenome.net/cas-offinder/). We found that the introduction of these mutations did not compromise the superior targeting specificity of HF1-Cas9 (Fig. EV3B). We also observed a slight increase in indel ratios ($p = 0.0382$, two-tailed, unpaired $t$-test), which is expected due to the overall increase in enzyme activity. This slight trade-off is a common phenomenon when optimizing genome editing tools. Further analysis of the editing window revealed that our variant did not alter the range of the editing window. (Appendix Fig. S10) These findings confirm the effectiveness of incorporating efficient mutation combinations into HF1-Cas9, further optimizing the safety of high-efficiency base editing. Overall, these results validate the versatility and usefulness of our one-size-fits-all engineering strategy across diverse base editor platforms.

## Mechanistic investigation and off-target analysis of engineered Cas9 variants

AI-8.3 features eight-point mutations, including four substitutions to positively charged amino acids. Given DNA's negative charge and previous findings showing that such mutations could enhance DNA binding affinity and improve editing efficiency, we propose that the enhanced editing efficiency of AI-8.3 might similarly result from its strengthened interaction with DNA (Nishimasu et al, 2014). Therefore, we first verified whether the mutated Cas9 protein enhanced binding affinity to DNA targets. We employed a CRISPRa fluorescence reporter system (Fig. 4A) and introduced the eight mutations into dCas9-VPR (with both RuvC and HNH nuclease catalytic sites deactivated). Additionally, we constructed

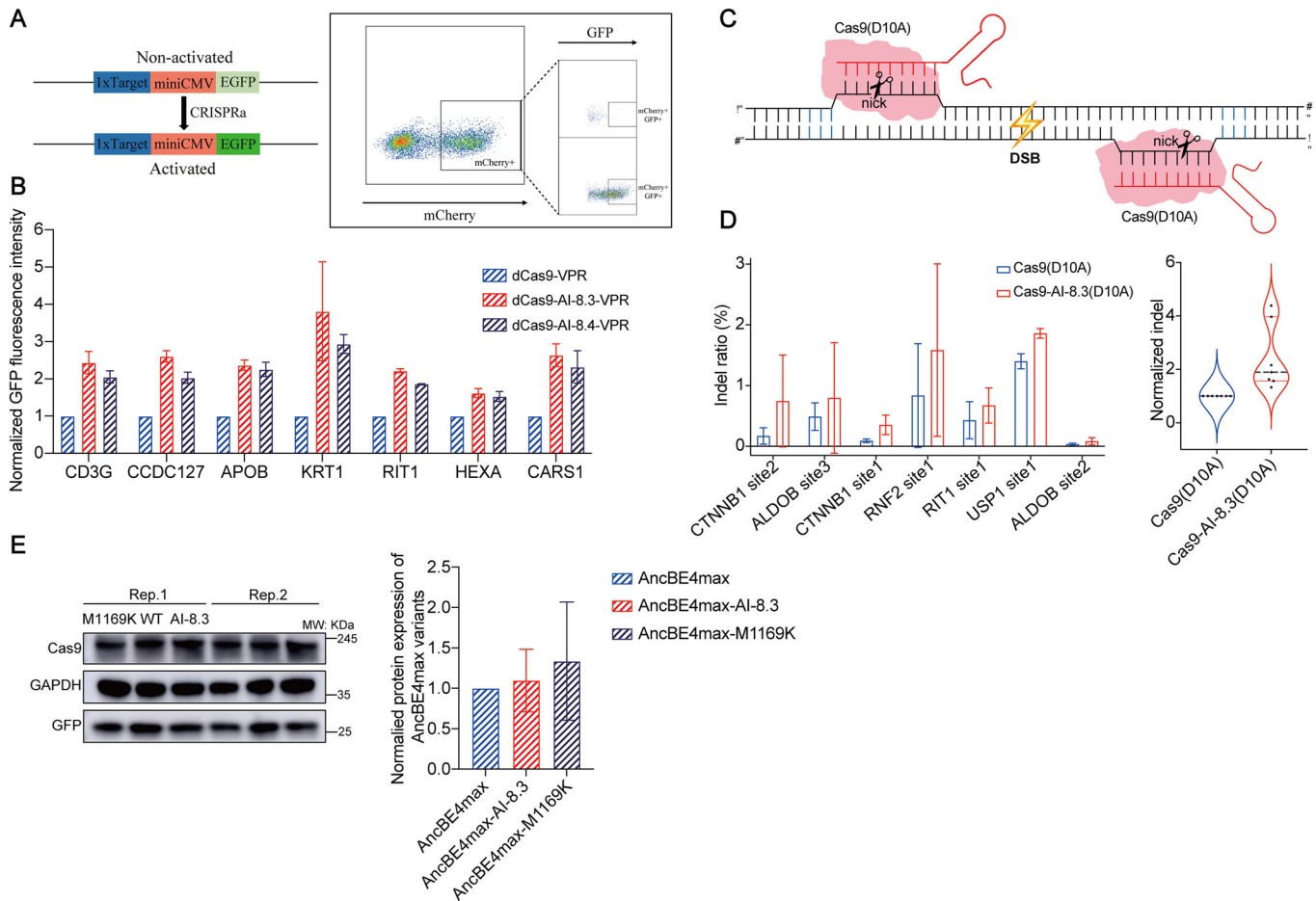

**Figure 4.  Mechanistic investigation of engineered Cas9 variants.**

(A) Schematic diagram of the CRISPRa assay. EGFP reporters are activated when targeted by the corresponding sgRNA. On the right, we present a representative flow cytometry result. (B) Normalized GFP fluorescence intensity of dCas9-VPR, dCas9-AI-8.3-VPR and dCas9-AI-8.4-VPR for each site shown in S10 ($n = 3$ biological replicates). (C) Cas9 (D10A) mediates double-strand cleavage under the guidance of two sgRNAs targeting complementary strands. (D) The indel ratio (%) of Cas9 (D10A) and Cas9-AI-8.3 (D10A) at seven genomic sites ($n = 3$ biological replicates). The right panel indicated the summary of the Indel ratio. The indel rates (%) induced by Cas9 (D10A) were set as 1. (E) Western blot results about the protein expression level of AncBE4max, AncBE4max-AI-8.3 and AncBE4max-M1169K are presented on the left. The right panel showed the quantitated results, the expression level of the AncBE4max was set as 1 ($n = 2$ biological replicates). Data information: In (B, D), data were presented as mean ± s.d. Source data are available online for this figure.

seven fluorescence reporter plasmids by inserting specific target sequences upstream of the miniCMV-EGFP cassette. The non-mutated or mutated versions of dCas9-VPR were co-transfected into HEK293T cells along with each reporter plasmid and their corresponding sgRNAs, followed by flow cytometry analysis to detect fluorescence intensity. Here, we selected two eight-mutation combinations (AI-8.3/8.4) for experimental validation, which shared identical mutation positions but differed only in the amino acid type at position 723 (Appendix Table S1), being isoleucine (8.3) and leucine (8.4), respectively. Results showed that dCas9-AI-8.3/8.4-VPR exhibited significantly increased fluorescence intensity (Fig. 4B; Appendix Fig. S9), demonstrating that our engineered variants enhanced the binding capability of Cas9 protein to DNA targets. Then we performed a molecular dynamics simulation. The results show that Cas9-AI-8.3 exhibits more stable dynamics compared to the wildtype, suggesting enhanced structural stability of Cas9-AI-8.3 (Appendix Fig. S12).

Since the third-generation base editor (BE3) replaced the dead version of Cas9 with nickase Cas9 (D10A), most base editors have maintained the single-strand nicking capability that promotes eukaryotic mismatch repair (MMR) or base excision repair (BER) of the unedited strand to improve the editing efficiency (Komor et al, 2016). This single-strand cleavage ability also reflects the activity of the Cas9 protein. Therefore, we introduced two sgRNAs targeting complementary strands at seven genomic sites and performed targeted nicking using either wild-type nickase Cas9 or its engineered variant nickase Cas9-AI-8.3 (Fig. 4C). The induced indel ratios were analyzed by NGS. Higher cleavage rates were observed in the nCas9-AI-8.3 group (Fig. 4D; Appendix Fig. S11), demonstrating that our engineering strategy directly enhanced Cas9 activity.

Since base editor expression also affects editing efficiency, we compared the protein expression levels of the original AncBE4max, AncBE4max-AI-8.3, and the M1169K mutant (which showed lower

editing activity than the WT version in previous tests, Fig. 1C) in HEK293T cells under identical plasmid amounts and transfection conditions (Fig. 4E). After normalization, it was observed that the protein level of AncBE4max-AI-8.3 showed no significant change compared to the WT group (approximately 1.1-fold), indicating that the enhanced editing performance is not related to increased protein expression.

DNA off-target activity is a vital concern for genome editors. Therefore, we investigated whether our Cas9-guided engineering strategy would increase Cas9-dependent or independent off-target rates. Using the Cas-OFFinder website (http://www.rgenome.net/cas-offinder/) (Bae et al, 2014), we predicted ten potential off-target sites (OT1-10) for two endogenous genomic loci, CTNNB1 and USP1. Through NGS analysis, we compared the base editing levels at predicted off-target sites between the wildtype and AI-AncBE4max groups (Fig. EV4A), which exhibited similar levels of off-target editing. Subsequently, we characterized the Cas9-independent off-target activity of the variant through orthogonal R-loop assays at six dead-enAsCas12f R-loop sites. Consistent with previous results, our high-performance variant showed similar off-target activity compared with the AncBE4max editor (Fig. EV4B). In sum, AI-AncBE4max improved editing efficiency without significantly increasing the risk of base editing off-target effects.

## Application of AI-guided high-efficiency base editor in human embryonic stem cells and multiple cancer cell lines

The ability to precisely and effectively edit genomic DNA sequences in living cells has long been a major goal in life sciences, especially for in vitro research. The rapid development of gene editing technology has opened new avenues for exploring gene function and treating diseases (Rees and Liu, 2018). As a novel gene editing technology that enables single-base substitutions without double-strand breaks, base editing has become a powerful tool in cancer treatment, drug screening, and regenerative medicine (Komor et al, 2016). The potential of hESCs to differentiate into various cell types holds great promise for regenerative medicine (Wobus and Boheler, 2005). Through base editing, we could not only target pathogenic mutations at early developmental stages, but also introduce disease-associated variants and establish disease models for studying therapeutic mechanisms (Shi et al, 2017; Wu et al, 2019). Moreover, base editing tools have particularly significant implications in cancer research. According to statistics from the ClinVar database, point mutations account for more than half of human pathogenic mutations (Fig. 5A), and the occurrence and development of cancer are more closely related to single-base mutations (Landrum et al, 2016).

To explore the potential applications of our AI-generated base editing tools, we selected hESCs and various cancer cell lines to compare the editing performance of AI-AncBE4max with the wild-type across multiple target sites. We selected seven high-incidence cancer cell lines derived from gastric cancer, non-small cell lung cancer, colorectal adenocarcinoma, liver cancer, and osteosarcoma. SgRNAs and base editors were co-transfected into these cancer cell lines, with mCherry fluorescence serving as a marker for successful transfection. Approximately 48 h post-transfection, the top 15% of mCherry-positive cells were collected using flow cytometry. Genomic DNA was then extracted, and NGS was employed to assess the editing performance between AI-AncBE4max and the

wild-type across the seven cancer cell lines. NGS results showed that AI-AncBE4max exhibited enhanced editing efficiency across all sites (Fig. 5B). Despite potential influences from transfection efficiency and inherent cellular characteristics that led to different baseline editing efficiencies among these cell lines, our AI multiple-mutant consistently maintained 1.4–2.1-fold higher average editing efficiency (Fig. 5C). Specifically, our editor showed the most pronounced efficiency improvements in HCT116 and U2OS cell lines, achieving more than 2.5-fold and even fivefold higher activity at certain sites. In AGS, HGC, PC-9, Caco-2, and HepG2 cell lines, AI-AncBE4max demonstrated average efficiency improvements ranging from 40 to 70%. We then evaluated the editing performance of AI-AncBE4max in hESCs across eight genomic loci (Fig. 5D). AI-AncBE4max demonstrated higher C-to-T conversion rates at all target sites, with an average increase of 1.6-fold over the wild-type (Fig. 5E). The indel frequencies for all cell lines are presented in Appendix Fig. S13, with mean rates consistently maintained at low levels (<1%) ($p > 0.05$, two-tailed, unpaired $T$-test). Overall, the comparison of editing efficiencies in both cancer cell lines and hESCs suggests that our AI-generated base editing tools offer great potential for applications in gene therapy and translational medicine, surpassing the commonly used base editors.

## Discussion

Since the development of CRISPR/Cas editing tools, base editors have played a crucial role in genetic engineering, particularly in the exploration and treatment of genomic diseases (Komor et al, 2016; Rees and Liu, 2018), though their editing efficiency and base-level accuracy remain modest. Previous studies have made various efforts to enhance base editing efficiency, such as introducing directed mutations into deaminases or Cas9, optimizing linker regions, incorporating single-strand binding proteins, and modifying nuclear localization signals (Kleinstiver et al, 2016; Koblan et al, 2018; Slaymaker et al, 2016; Wang et al, 2017). However, there remains considerable room for improving the efficiency of base editors, especially cytosine base editors. Meanwhile, advances in artificial intelligence technology have opened new opportunities for protein engineering (Abramson et al, 2024; AlQuraishi, 2021; Jumper et al, 2021; Madani et al, 2023; Notin et al, 2024; Riesselman et al, 2018; Senior et al, 2020; Yeh et al, 2023). Therefore, we applied our protein mutation effect predictor (ProMEP) to develop an AI-guided Cas9 engineering strategy and attempted to improve base editing performance for various versions of base editors (Cheng et al, 2024).

We first used ProMEP to predict potentially beneficial mutations in the Cas9 protein. We observed significant enrichment of X-to-K mutants in the top 5% of all ranked mutants. We attribute this phenomenon to prediction biases arising from imbalanced amino acid distributions in natural Cas9 sequences. Using 123 Cas9 homologs (sequence identity <70%) from the non-redundant (NR) protein sequence database, we found that lysine (K) accounts for 10.94% of residues (vs. arginine [R] = 5.11%, histidine [H] = 2.01%, aspartate [D] = 6.51%). As ProMEP was pretrained on ~160 million proteins, it internalizes evolutionary biases favoring naturally abundant residues. When predicting fitness scores for Cas9 mutants through sequence-structure context embedding, the

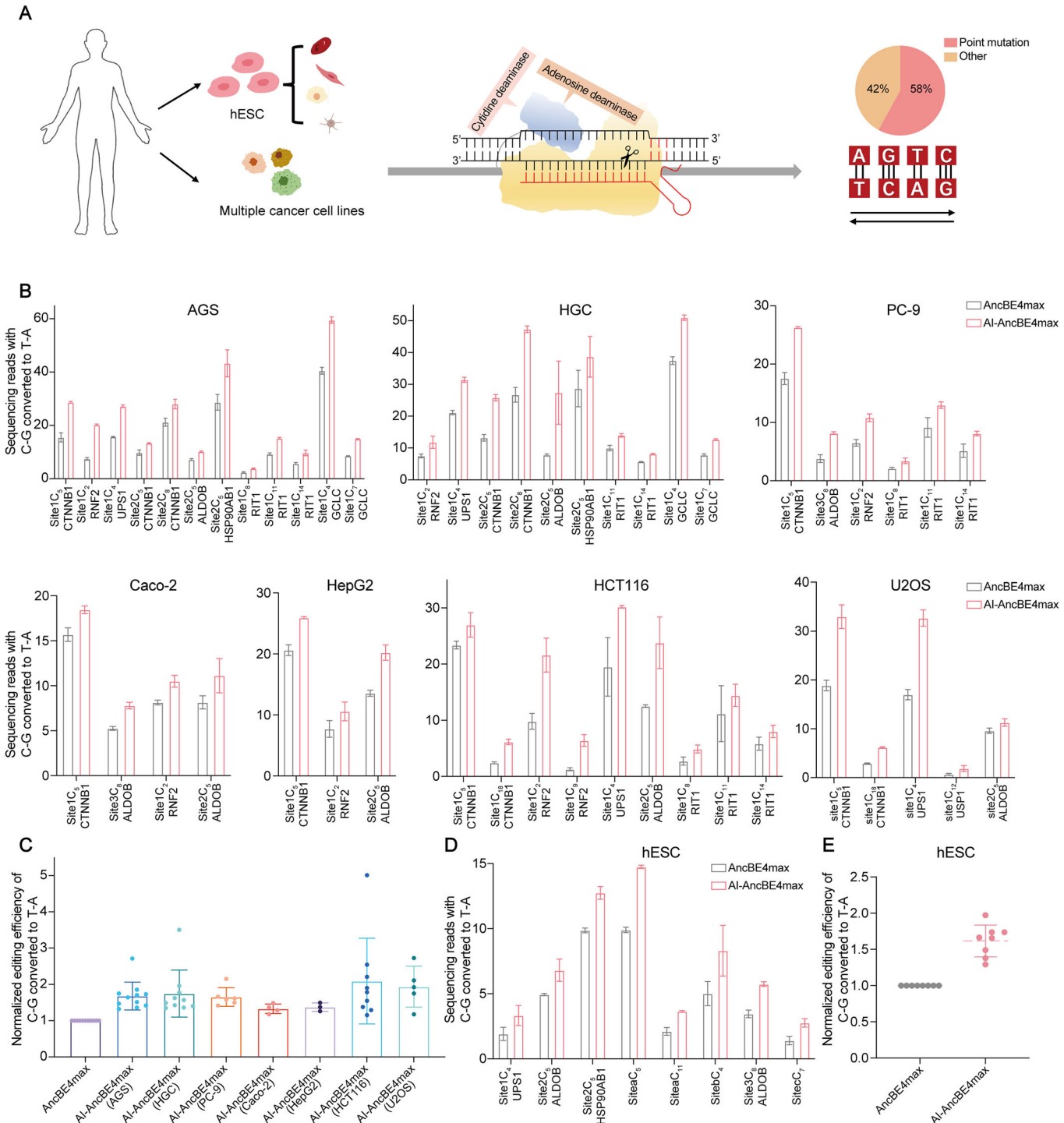

**Figure 5. Application of an AI-guided high-efficiency base editor.**

(A) The pie chart represents the distribution of human pathogenic genetic variants, where point mutations constitute 58%. Base editors can perform various types of base conversions in human embryonic stem cells and diverse cancer cell lines, demonstrating their powerful application potential. (B) Editing efficiencies of AI-AncBE4max and AncBE4max at all editing positions across several genomic sites in AGS, HGC, PC-9, Caco-2, HepG2, HCT116, and U2OS cells ($n = 3$ biological replicates). (C) Normalized editing efficiencies for all sites shown in (B) ($n = 12, 10, 6, 4, 3, 9,$ and 5, respectively). The editing level of AncBE4max at each site were set as 1. (D) Sequencing reads with C-to-T of AncBE4max and AI-AncBE4max at all editing positions across seven sites in hESCs ($n = 3$ biological replicates). (E) Normalized C-to-T conversion (%) shown in (D) ($n = 8$). Data information: In (B–E), data were presented as mean ± s.d. Source data are available online for this figure.

observed K-enrichment in Cas9 homologs may predispose ProMEP to favor X-to-K mutations. Notably, ProMEP still identified diverse beneficial substitutions (e.g., G1218R, T622W, V1015K, and C80M), demonstrating its capacity to transcend evolutionary biases while leveraging conserved fitness patterns.

We then used ProMEP to select 18 high-ranking mutations for experimental validation via a hybrid strategy. Among these mutations, G1218R demonstrated the highest performance, with 1.64-fold higher efficiency compared to WT. Evaluation of mutation effect prediction performance on both the Cas9 deep mutation scanning dataset and our experimental dataset further confirms ProMEP's effectiveness (Data ref: (Notin et al, 2023)) (Appendix Fig. S14, Methods). Subsequently, we combined G1218R with several other single-point mutations that outperformed WT to create Cas9-AI-5. Beyond simply combining beneficial mutations, we also considered potential synergistic effects. Therefore, we matched Cas9-AI-5 with saturation mutation libraries at other positions and used ProMEP to calculate fitness scores for each multi-point mutant, testing the top ten eight-point combinations. This screening process led to the identification of AncBE4max-AI-8.3, which exhibited an average efficiency 2.34-fold higher than AncBE4max and was subsequently designated as AI-AncBE4max. Our strategy was then extended to other base editor variants, where we observed consistent efficiency improvements across diverse systems. These results underline the versatility and broad applicability of the AI-driven engineering approach. In particular, the development of AI-HF-BE allowed us to achieve a base editor with both high editing efficiency and low off-target effects, overcoming the long-standing trade-off between efficacy and safety in base editing. This is critical for therapeutic genome editing and represents a key advancement toward safe in vivo applications.

To investigate potential mechanisms of the enhanced efficiency, we conducted CRISPRa activation assays, molecular dynamics simulation, nCas9 cleavage experiments, and quantitative analysis of protein expression levels. The results suggested that our engineering strategy improved both the DNA target binding capability and structural stability of the Cas9 protein, and enhanced its intrinsic cleavage activity, while showing no correlation with protein expression levels. Considering the safety aspects of tool application, we evaluated off-target editing levels associated with both sequence similarity and R-loop formation. The results revealed that AI-AncBE4max maintained off-target rates similar to those of AncBE4max.

To explore the application potential of AI-AncBE4max, we assessed the performance of this engineered variant across multiple high-incidence cancer cell lines and human embryonic stem cells. The relevant data showed that the editing activity of AI-AncBE4max is significantly higher than that of the control versions, while consistently maintaining a lower indel rate. Although ProMEP can currently predict the effects of any amino acid substitution in protein sequences, this strategy appears to only influence editing efficiency without altering other characteristics of base editors, such as editing window and sequence preference. Development of prediction models for different purposes may require larger training datasets. Overall, in this study, we validated that AI models can serve as a more comprehensive and accurate protein engineering tool, and the resulting high-performance Cas9 variant provides a universal improvement strategy for a series of gene editing tools.

## Methods

### Reagents and tools table

| Reagent/resource | Reference or source | Identifier or catalog number |
|---|---|---|
| **Experimental models** | | |
| HEK293T cells | ATCC | CRL-3216 |
| HeLa | ATCC | CCL-2 |
| AGS | ATCC | CRL-1739 |
| Caco-2 | ATCC | HTB-37 |
| HepG2 | ATCC | HB-8065 |
| HCT116 | ATCC | CCL-247 |
| U-2 OS | ATCC | HTB-96 |
| **Recombinant DNA** | | |
| AncBE4max | Addgene | #112094 |
| YEE-BE4max | Addgene | #138157 |
| CGBE | Addgene | #163552 |
| ABE-max | Addgene | #112095 |
| ABE8e | Addgene | #138495 |
| pCMV-SpCas9 | Addgene | #139987 |
| pGL3-U6-sgRNA-mCherry | Our lab | |
| **Antibodies** | | |
| anti-Cas9 | ABclonal | A14997 |
| anti-GAPDH | ABclonal | AC027 |
| anti-GFP | ABclonal | AE011 |
| Goat anti-rabbit IgG | Thermo | 32460 |
| **Oligonucleotides and other sequence-based reagents** | | |
| PCR primers | This study | Appendix Tables S2/3/4 |
| **Chemicals, enzymes and other reagents** | | |
| 2 × Phanta Flash Master Mix [Dye Plus] | Vazyme | P505 |
| BasI-HFv2 | NEB | R3733 |
| ClonExpress II One Step Cloning Kit | Vazyme | C115 |
| DNA Ligase Solution I | Takara | 6022Q |
| Dulbecco's modified Eagle's medium | Gibco | 12491015 |
| RPMI 1640 Medium | Gibco | 12633020 |
| FBS | Gemini | 100-106 |
| penicillin streptomycin | Gibco | 15140-122 |
| EZ Trans | Life-iLab | AC04L091 |
| mTeSR1 | STEMCELL | 85850 |
| Matrigel | BioCoat | 354277 |
| Accutase | STEMCELL | 07920 |
| Gentle Cell Dissociation Reagent cGMP | STEMCELL | 100-0485 |
| Y-27632 | STEMCELL | 100-1044 |
| Lipofectamine Stem reagent | Thermo | STEM00015 |

| Reagent/resource | Reference or source | Identifier or catalog number |
|---|---|---|
| QuickExtract™ DNA Extraction Solution | Lucigen | QE09050 |
| E. coli cells | Sangon | B528413 |
| **Software** | | |
| FlowJo v10 | | https://www.flowjo.com/flowjo10/download |
| CRISPResso2 | | https://crispresso.pinellolab.org/submission |
| Cas-OFFinder | | http://www.rgenome.net/cas-offinder |
| ImageJ | | https://imagej.nih.gov/ij/index.html |
| GraphPad Prism 9 | | https://www.graphpad.com |
| Snapgene | | https://www.snapgene.com/ |
| **Other** | | |
| Illumina NovaSeq (PE150) | | Novogene Technology (Beijing, China) |
| ProMEP | | https://github.com/wenjiegroup/ProMEP |

## ProMEP-guided protein engineering of Cas9

In our previous work (Cheng et al, 2024), we have evaluated the performance of ProMEP on a series of benchmarks, including the ProteinGYM benchmark, the pathogenicity benchmark, and the stability benchmark. Specifically, the ProteinGym benchmark covers 53 proteins and 1.43 million variants with experimentally determined function data. Leading mutation effect prediction methods were evaluated for comparison, including AlphaMissense (Cheng et al, 2023) and ESM2-650M (Lin et al, 2023). On the ProteinGYM benchmark, ProMEP achieves an average Spearman's rank correlation of 0.523, significantly outperforming the best protein language model, ESM2-650M (average Spearman's rank correlation of 0.465, P value = 0.03; t-test, two-sided). Besides, ProMEP demonstrates two to three orders of magnitude speed improvement with comparable performance to the best multimodal method, AlphaMissense (average Spearman's rank correlation of 0.523 vs 0.520, P value = 0.91; t-test, two-sided). In the pathogenicity benchmark and stability benchmark, ProMEP consistently demonstrates state-of-the-art performance.

ProMEP integrates both protein sequence and structural contexts to predict the fitness score of protein variants. As demonstrated in the equation, ProMEP employs the sequence context $S$ and the structure context $C$ as inputs, generating a matrix of log probabilities that delineates the distribution of the 20 amino acids across various positions within a protein. For a given variant, ProMEP computes the conditional probabilities of the mutated amino acid $mt$ and the wild-type amino acid $wt$ at each mutational position $t$. The summation over all mutated positions $T$ yields the final fitness score of the specific protein variant.

$$\sum_{t \in T} \log p(x_t = x_t^{mt}|S_{-t}, C) - \log p(x_t = x_t^{wt}|S_{-t}, C)$$

During Cas9 engineering, we first calculated the normalized fitness score of all single mutants and ranked them accordingly. We then analyzed the mutation-type enrichment in the top 5% of these ranked mutants. To transcend evolutionary biases while leveraging conserved fitness patterns, we selected 18 high-fitness single mutants based on the enrichment analysis, comprising: the top 6 X-to-K mutants, the top 3 X-to-C mutants, the top 3 X-to-W mutants, the top 3 X-to-M mutants and the top 3 X-to-R mutants. For triple mutants, we selected the top 10 candidates based on normalized fitness scores.

## Evaluation of mutation effect prediction performance on Cas9 datasets

We evaluated ProMEP's mutation effect prediction performance on two datasets: the ProteinGym Cas9 benchmark (Cas9_ProteinGym, containing 8117 single-point mutants) and an experimental dataset (Cas9_18mutants, containing 18 single-point mutants). Notably, five mutants (C80R, C80W, G1218R, M1169K, and M1169R) overlap between datasets. Current leading methods, including ESM2-650M, ESM2-3B, Tranception (Large, no retrieval), and ProstT5, are used for comparison. While all methods showed comparable performance on Cas9_ProteinGym (Figure X), ProMEP achieves a Spearman's ρ of 0.736 on Cas9_18mutants, outperforming ESM2-650M (ρ = 0.404) and Tranception (ρ = 0.251). We attribute ProMEP's performance on Cas9_18mutants to the hybrid strategy combining fitness score thresholds with mutation-type quotas. The small sample size of Cas9_18 mutants may also contribute to performance variability (Data ref: (Notin et al, 2023)).

## Plasmid construction

pCMV-SpCas9, AncBE4max, YEE-BE4max, CGBE, ABE-max, and ABE8e plasmids were purchased from Addgene (Addgene, #139987, #112094, #138157, #163552, #112095, and #138495). Engineered BE variants, nCas9 and dCas9-VPR were constructed by circular PCR using specific primers containing the desired variants and the respective plasmids as templates using 2 × Phanta Flash Master Mix [Dye Plus] (Vazyme). The sequence information for circular PCR are displayed in Appendix Table S2. For the sgRNA, the plasmid backbone pGL3-U6-sgRNA-mCherry was cut by BasI-HFv2 (NEB) for overhangs. The spacer oligo with compatible overhangs (top strand with ends of 5'ACCG, bottom strand with 5'AAAC) was synthesized and annealed. The sequence information of sgRNA was provided in Appendix Table S3. Two components were assembled by DNA Ligase Solution I (DNA Ligation Kit Ver.2.1, Takara). The above PCR products and ligated plasmids were transformed into E. coli and selected using ampicillin.

## Cell culture, transfection, and cell sorting

HEK293T, HeLa, AGS, HGC, HCT116, Caco-2, pc-9, U2OS, and HepG2 cells were maintained in Dulbecco's modified Eagle's medium (Gibco), supplemented with 10% fetal bovine serum (FBS) (v/v) (Gemini) and 1% penicillin streptomycin (v/v) (Gibco). Cell lines were validated and tested for mycoplasma contamination periodically. Cells were seeded on poly-D-lysine-coated 24-well plates at 37 °C with 5% $CO_2$ and transfected at ~80% confluence using EZ Trans (Life-iLab, AC04L091) reagent with 1.2 ug plasmids (including 900 ng base editors or pCMV-Cas9 (D10A), 300 ng sgRNA) per well. H1 hESCs were cultured in mTeSR1 (STEMCELL Technologies) on Matrigel (STEMCELL Technologies)-coated

plates and dissociated with Accutase (Thermo). Before transfection, cells were plated with 10 μM Y-27632 (Tocris) in the precoated 48-well plate. A total of 700 ng plasmids (including 500 ng base editors and 200 ng sgRNA) were transfected into cells per well by using Lipofectamine Stem reagent (Thermo) according to the manufacturer's protocols. After 48 h of transfection, mCherry+ cells were harvested by Fluorescence-activated cell sorting (FACS).

### Genomic DNA extraction and target deep sequencing

Genomic DNA of the harvested cells was extracted with the aid of QuickExtract™ DNA Extraction Solution (Lucigen) according to the manufacturer's protocols and used as templates for PCR amplification with 2 × Phanta Flash Master Mix [Dye Plus] (Vazyme). Primers used were listed in Appendix Table S4. For deep sequencing, we added 5-nt barcodes to the 5' end of forward primers for PCR amplification and used the Illumina NovaSeq (PE150) at the Novogene Technology (Beijing, China). Clean reads were obtained from the company, and CRISPResso2 was used for editing efficiency analysis. Base editing efficiency was calculated with the BE mode, and Indel frequency was calculated using the NHEJ mode.

### Off-target analysis

For Cas9-dependent off-target analysis, the Cas-OFFinder tool was used (http://www.rgenome.net/cas-offinder/) to predict potential off-target sites. To analyze Cas9-independent off-target effects, we used dead-enAsCas12f and corresponding sgRNAs to generate R-loops at specific genomic sites. The sequences around the predicted sites and the R-loop sites were amplified using 2 × Phanta Flash Master Mix [Dye Plus] (Vazyme) and subjected to deep sequencing using the Illumina NovaSeq (PE150) at the Novogene Technology (Beijing, China). Editing efficiency was analyzed by the same methods mentioned in the previous section.

### CRISPRa assay

We did transfection in HEK293T cells with 500 ng CMV-dCas9-VPR, 100 ng targeted EGFP reporter and 300 ng sgRNA (containing a mCherry marker) using EZ trans according to the manufacturer's protocols. Cells were analyzed using flow cytometry after 48 h. CRISPRa activity was assessed using normalized reporter expression, defined as the ratio of cells displaying equivalent high EGFP fluorescence to the total mCherry-positive cell population. The results were analyzed by FlowJo (v10).

### Western blot

RIPA lysis buffer was used to obtain total proteins from HEK293T cells after 48 h of transfection. The primary antibodies used included anti-Cas9 (ABclonal A14997, 1:1000), anti-GAPDH (ABclonal AC027, 1:5000), and anti-GFP (ABclonal AE011, 1:2000). Goat anti-rabbit IgG (Thermo 32460, 1:5000) was used as the secondary antibody. Images were captured with Amersham Imager 600 and analyzed for grayscale using ImageJ software.

### Statistical analyses

All the data in this study were based on three biological replicates, except for Western blot data. GraphPad Prism (version 9) was used for graphing and statistical analyses. Results are presented as means ± SD as indicated in the figure legends. Statistical analysis was performed with a two-tailed unpaired $t$-test.

No blinding was performed in these studies.

## Data availability

The datasets and computer code produced in this study are available in the following databases: Deep-sequence data: National Center for Biotechnology Information Sequence Read Archive PRJNA1192552. Modeling computer scripts: GitHub (https://github.com/wenjiegroup/ProMEP).

The source data of this paper are collected in the following database record: biostudies:S-SCDT-10_1038-S44320-025-00142-0.

## Peer review information

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

## Acknowledgements

We would like to thank members of Wei lab, Huang lab, and Shu lab for their contributions in providing experimental materials and helpful discussions. We thank the Molecular Imaging Core Facility (MICF) at the School of Life Science and Technology, ShanghaiTech University, for providing technical support. This work is supported by the National Natural Science Foundation of China [12331018 and 32401220], local grant [CXPJJH122006-1014], Capacity Building Project of the Key Technology Platform, SIBPT [PT2025-03-01] and Innovation Promotion Program of NHC and Shanghai Key Labs, SIBPT [RC2025-01]. And we thank all collaborators who kindly donated cell lines.

## Author contributions

**Dongyi Wei**: Data curation; Formal analysis; Validation; Visualization; Writing—original draft; Writing—review and editing. **Peng Cheng**: Data curation; Software; Formal analysis; Investigation; Visualization. **Ziguo Song**: Software; Formal analysis; Methodology; Writing—original draft. **Yixin Liu**: Resources. **Xiaoran Xu**: Resources. **Xingxu Huang**: Resources; Supervision; Project administration. **Xiaolong Wang**: Resources; Supervision. **Yu Zhang**: Resources; Supervision; Funding acquisition; Project administration; Writing—review and editing. **Wenjie Shu**: Resources; Software; Supervision; Funding acquisition; Project administration. **Yongchang Wei**: Conceptualization; Resources; Supervision; Funding acquisition; Project administration; Writing—review and editing.

Source data underlying figure panels in this paper may have individual authorship assigned. Where available, figure panel/source data authorship is listed in the following database record: biostudies:S-SCDT-10_1038-S44320-025-00142-0.

## Disclosure and competing interests statement

The authors declare no competing interests.

# Expanded View Figures

## A

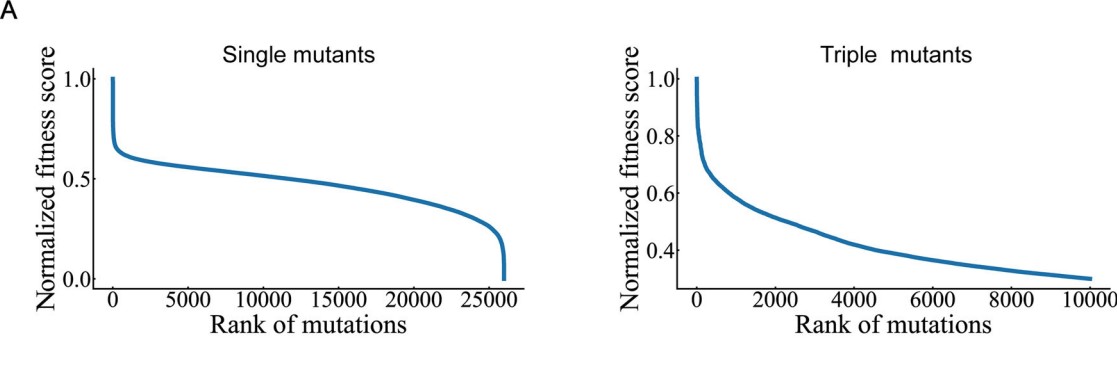

## B

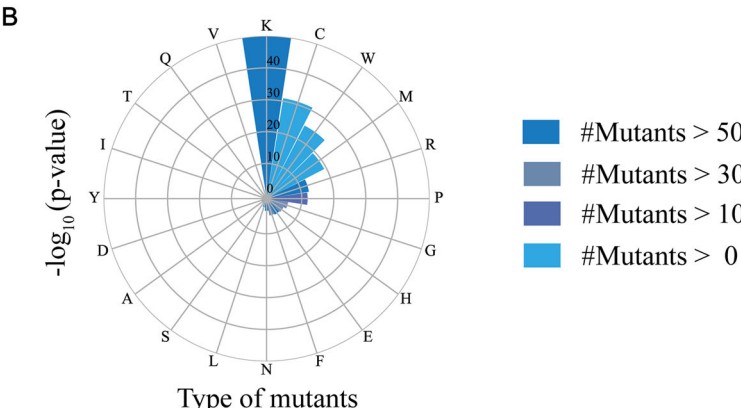

**Figure EV1. Enrichment analysis of top-ranked mutants.**

(**A**) Normalized fitness score predicted by ProMEP. (**B**) Each polar axis delineates a specific category of mutants (e.g., the K axis denotes X-to-K mutants, while the C axis signifies X-to-C mutants). The radial tick marks correspond to the $-\log_{10}(p$ value) derived from the enrichment analysis results. The coloration of each bar indicates the count of a particular type of mutant (e.g., among the top 5% mutants, there are 264 X-to-K mutants).

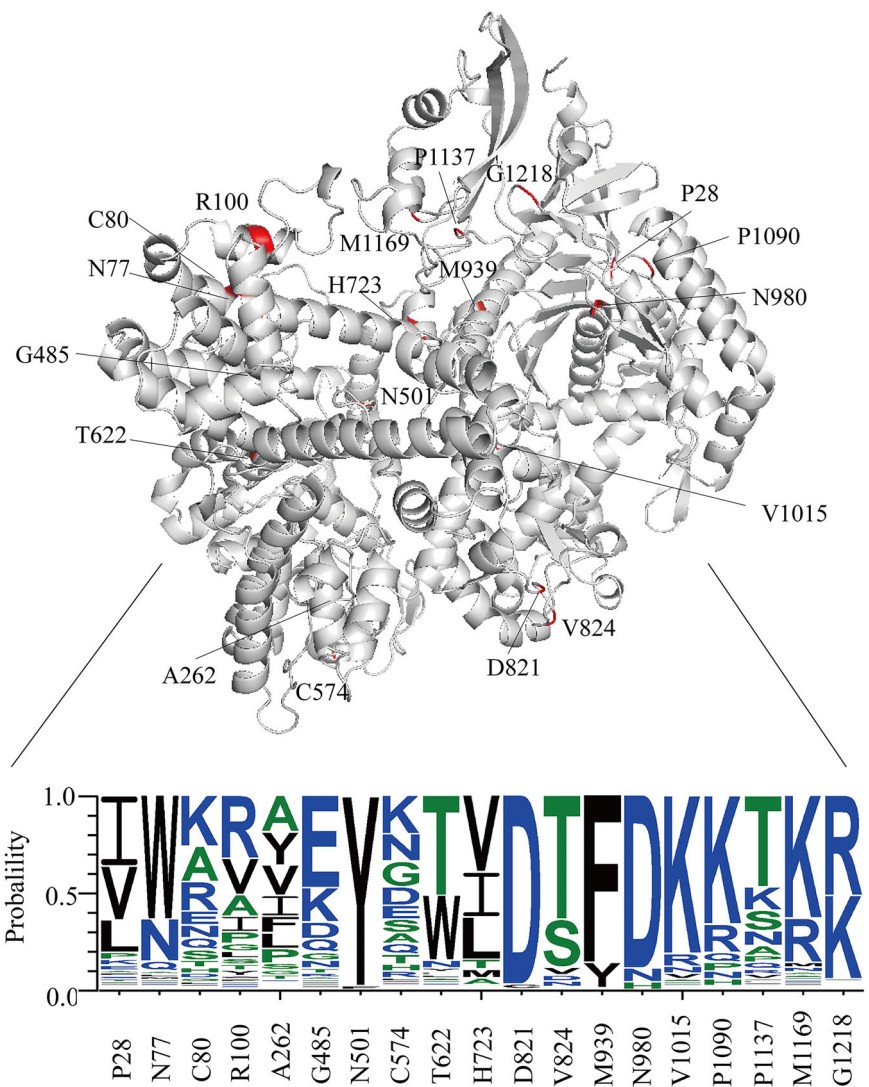

**Figure EV2. Probability distribution of Cas9.**

The predicted probability distributions of all amino acid types at altered positions within Cas9.

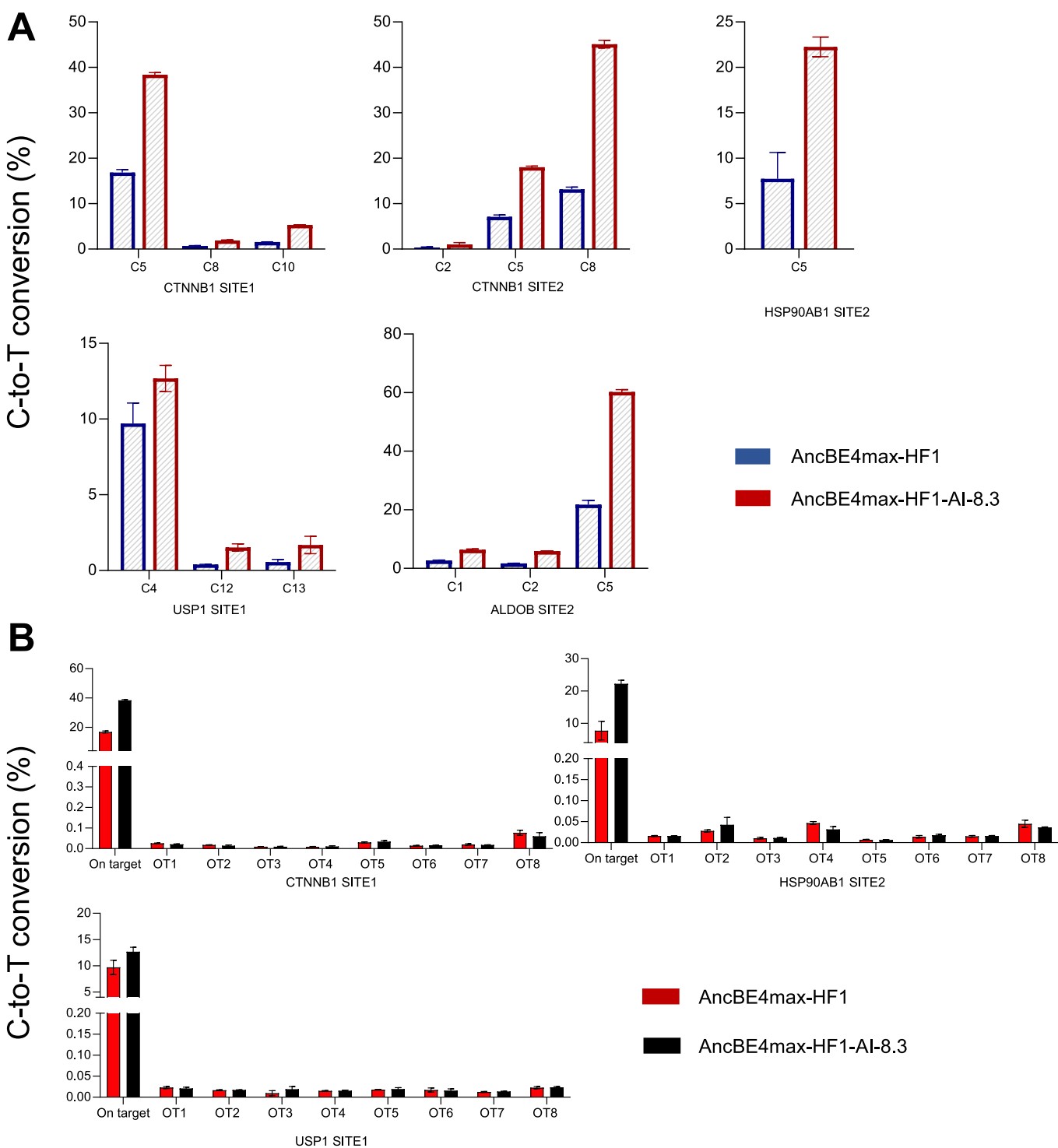

**Figure EV3. Editing efficiency and Cas9-dependent off-target effect of HF1-CBEs.**

(A) The editing efficiency of AncBE4max-HF1 and AncBE4max-HF1-AI-8.3 at all editing positions across five sites in HEK293T cells ($n = 3$ biological replicates). (B) Base editing levels for on-target sites (CTNNB1, USP1, and HSP90AB1) and eight corresponding predicted off-target sites were compared between AncBE4max-HF1 wildtype and AI-8.3 groups through NGS analysis ($n = 3$ biological replicates). Data information: data were presented as mean ± s.d.

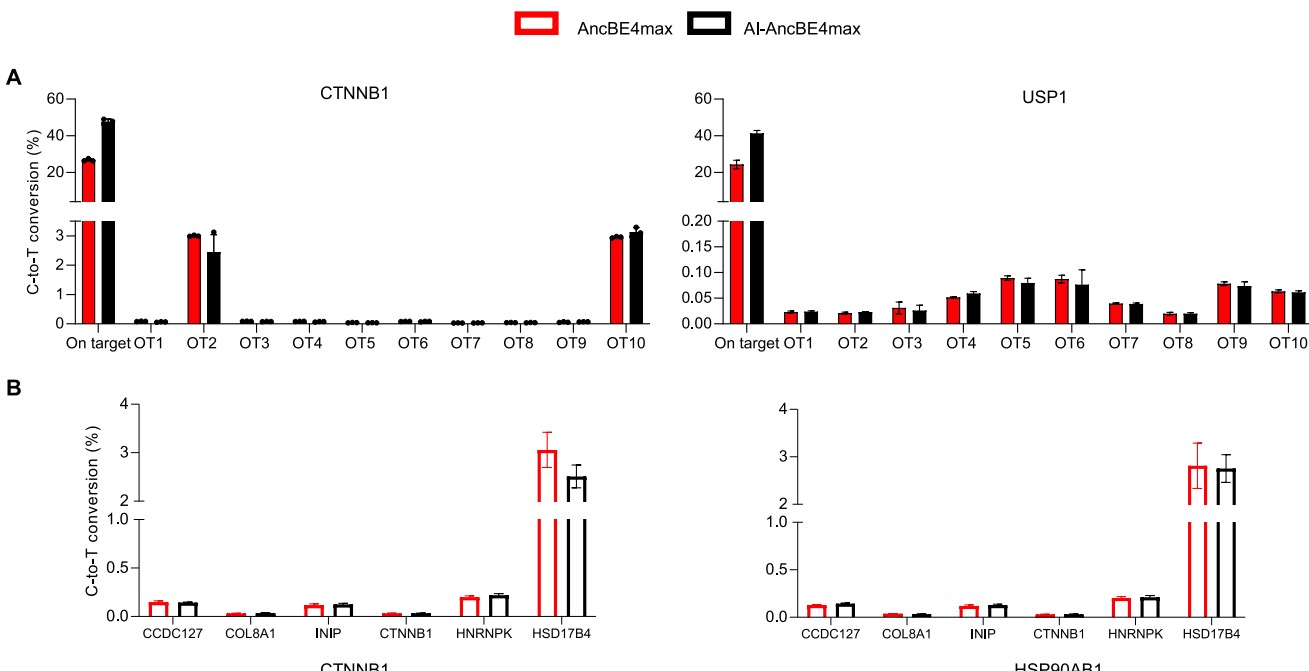

**Figure EV4.   Off-target analysis of AncBE4max and AI-AncBE4max.**

(A) Cas9-dependent off-target analysis. Base editing levels for on-target sites (CTNNB1 and USP1) and ten corresponding predicted off-target sites were compared between wildtype and AI-AncBE4max groups through NGS analysis ($n = 3$ biological replicates). (B) Cas9-independent off-target analysis. Base editing levels of six dead-enAsCas12f R-loop sites at two Cas9 targets (CTNNB1 and HSP90AB1) ($n = 3$ biological replicates). Data information: data were presented as mean ± s.d.

