## [Peer Review File · Molecular Systems Biology]

AI-guided Cas9 engineering provides an effective strategy to enhance base editing

Dongyi Wei, Peng Cheng, Ziguang Song, Yixin Liu, Xiaoran Xu, Xingxu Huang, Xiaolong Wang, Yu Zhang, Wenjie Shu, and Yongchang Wei

Corresponding author(s): Yongchang Wei (weiyongchang@whu.edu.cn), Wenjie Shu (shuwj@bmi.ac.cn), Yu Zhang (zhangy@sibpt.cn)

Review Timeline:

Submission Date:	3rd Feb 25
Editorial Decision:	10th Apr 25
Revision Received:	8th Jul 25
Editorial Decision:	8th Aug 25
Revision Received:	15th Aug 25
Accepted:	21st Aug 25

Editor: Poonam Bheda

Transaction Report:

10th Apr 2025

Manuscript Number: MSB-2025-12897-T

Title: AI-guided Cas9 engineering provides an effective strategy to enhance base editing

Dear Prof. Wei,

Thank you for the submission of your manuscript to Molecular Systems Biology. We have now received feedback from the two/three reviewers who agreed to evaluate your manuscript. As you will see from the reports below, the referees acknowledge the interest of the study and are overall supporting publication of your work pending appropriate revisions.

I think that the recommendations of the reviewers are rather clear and I therefore do not see the need to repeat the comments listed below. All issues raised would need to be satisfactorily addressed. Please let me know in case you would like to discuss in further detail any of the any of the reviewer comments or your proposed revisions, I would be happy to schedule a call.

We require:

1) A .docx formatted version of the manuscript text (including legends for main figures, EV figures and tables). Please make sure that the changes are highlighted to be clearly visible. Alternatively you may choose to submit your manuscript as a LaTeX file.

4) A .docx formatted letter INCLUDING the reviewers' reports and your detailed point-by-point responses to their comments. As part of the EMBO Press transparent editorial process, the point-by-point response is part of the Peer Review File (PRF), which will be published alongside your paper.

5) A complete author checklist, which you can download from our author guidelines (<https://www.embopress.org/page/journal/17574684/authorguide#submissionofrevisions>). Please insert information in the checklist that is also reflected in the manuscript. The completed author checklist will also be part of the PRF.

6) Please note that all corresponding authors are required to supply an ORCID ID for their name upon submission of a revised manuscript.

7) It is mandatory to include a 'Data Availability' section after the Materials and Methods. Before submitting your revision, primary datasets produced in this study need to be deposited in an appropriate public database, and the accession numbers and database listed under 'Data Availability'. Please remember to provide a reviewer password if the datasets are not yet public (see <https://www.embopress.org/page/journal/17574684/authorguide#dataavailability>).

In case you have no data that requires deposition in a public database, please state so in this section as follows: "This study includes no data deposited in external repositories". Note that the Data Availability Section is restricted to new primary data that are part of this study.

8) All Materials and Methods need to be described in the main text using our 'Structured Methods' format, which is required for all research articles. According to this format, the Methods section includes a Reagents and Tools Table (listing key reagents, experimental models, software and relevant equipment and including their sources and relevant identifiers) followed by a Methods and Protocols section describing the methods using a step-by-step protocol format. The aim is to facilitate adoption of the methodologies across labs. Please upload the Reagents and Tools table as a separate document when submitting your revised manuscript. More information on how to adhere to this format as well as a downloadable template (.docx) for the Reagents and Tools Table can be found in our author guidelines:

<https://www.embopress.org/page/journal/17444292/authorguide#structuredmethods>

9) For data quantification: please specify the name of the statistical test used to generate error bars and p-values, the number (n) of independent experiments (specify technical or biological replicates) underlying each data point and the test used to calculate p-values in each figure legend. The figure legends should contain a basic description of n, p-values and the test applied. Graphs must include a description of the bars and the error bars (s.d., s.e.m.). Please provide exact p-values (in either the figure or figure legend).

10) Our journal encourages inclusion of *data citations in the reference list* to directly cite datasets that were re-used and obtained from public databases. Data citations in the article text are distinct from normal bibliographical citations and should directly link to the database records from which the data can be accessed. In the main text, data citations are formatted as follows: "Data ref: Smith et al, 2001" or "Data ref: NCBI Sequence Read Archive PRJNA342805, 2017". In the Reference list, data citations must be labeled with "[DATASET]". A data reference must provide the database name, accession number/identifiers and a resolvable link to the landing page from which the data can be accessed at the end of the reference. Further instructions are available at .

11) We replaced Supplementary Information with Expanded View (EV) Figures and Tables that are collapsible/expandable online. EV Figures should be cited as 'Figure EV1, Figure EV2' etc... in the text and their respective legends should be included in the main text after the legends of regular figures.

- Additional Tables/Datasets should be labeled and referred to as Table EV1, Dataset EV1, etc. Legends should be provided in a separate tab in case of .xls files. Alternatively, the legend can be supplied as a separate text file (README) and zipped together with the Table/Dataset file.

<https://www.embopress.org/page/journal/17574684/authorguide#expandedview>

12) Author contributions: CRediT has replaced the traditional author contributions section because it offers a systematic machine-readable author contributions format that allows for more effective research assessment. Please remove the Authors Contributions from the manuscript and use the free text boxes beneath each contributing author's name in our system to add specific details on the author's contribution. More information is available in our guide to authors.

13) Disclosure statement and competing interests: We updated our journal's competing interests policy in January 2022 and request authors to consider both actual and perceived competing interests. Please review the policy <https://www.embopress.org/competing-interests> and update your competing interests if necessary.

14) Every published paper now includes a 'Synopsis' to further enhance discoverability. Synopses are displayed on the journal webpage and are freely accessible to all readers. They include a short stand first (maximum of 300 characters, including space) as well as 2-5 one-sentences bullet points that summarizes the paper. Please write the bullet points to summarize the key NEW findings. They should be designed to be complementary to the abstract - i.e. not repeat the same text. We encourage inclusion of key acronyms and quantitative information (maximum of 30 words / bullet point). Please use the passive voice. Please attach these in a separate file or send them by email, we will incorporate them accordingly.

Please note that these would be the final versions and changes during proofing are usually not allowed.

15) As part of the EMBO Publications transparent editorial process initiative (see our policy here:

https://www.embopress.org/transparent-process#Review_Process), Molecular Systems Biology will publish online a Peer Review File (PRF) to accompany accepted manuscripts.

In the event of acceptance, this file will be published in conjunction with your paper and will include the anonymous referee reports, your point-by-point response and all pertinent correspondence relating to the manuscript. Let us know whether you agree with the publication of the PRF and as here, if you want to remove or not any figures from it prior to publication.

Please note that the Author checklist will be published at the end of the PRF.

Molecular Systems Biology has a "scooping protection" policy, whereby similar findings that are published by others during review or revision are not a criterion for rejection. Should you decide to submit a revised version, I do ask that you get in touch after three months if you have not completed it, to update us on the status.

Yours sincerely,

Poonam Bheda, PhD
Scientific Editor

Reviewer #1:

This study presents an AI-guided engineering approach to enhance base editing efficiency by optimizing the Cas9 protein. The researchers used the ProMEP tool to predict beneficial mutations in Cas9 and successfully developed a high-performance variant, AncBE4max-AI-8.3, which showed a 2-3-fold increase in average editing efficiency. This engineered Cas9 also effective in improving the performance of various base editors across multiple cell lines, demonstrated that AI models could serve as effective protein engineering tools for gene editing applications.

The study provides convincing evidence that the AI-engineered Cas9 variant can significantly enhance base editing efficiency. The significance of the progress of the study lies in the fact that the use of language models not only predicts the activity of single mutations, but also gives better prediction results in combination mutations.

However, the article also has the following problems that need to be explained or modified.

1. The article is written clearly and coherently as a whole, but the method part does not explain the principle and accuracy of the ProMEP algorithm, and it is necessary to fill in the top scores of single mutation and combined mutation prediction in the attached table. There are many AI tools with few and zero samples published, why this method is used, and whether it has advantages over other tools?
2. The experience of many past studies in the field of Cas9 activity optimization is that the positive charge mutation of R or K at the key position can improve the activity, but the prediction results of the software in Figure EV1 are concentrated on K, and the proportion of R is very small, and the experimental results also prove that the most important V1015 mutation is much higher than the activity of R than K, how to explain such a prediction bias?
3. The study mainly focuses on the improvement of editing efficiency, and the experimental results showed that the off-target effect was similar to that of the wild type. However, the clinical application of wild-type SpCas9 is limited due to its high off-target rate, and SpCas9-HF, a low-off-target mutant, has long been studied. If the four REC domain mutations of SpCas9-HF1 are introduced in experimental, it is of practical significance to optimize the variants with lower off-target rate.
4. A minor points in part of result, AI-AncBE4max groups (Figure EV11) need to be changed to EV12.

Reviewer #2:

Wei et al. present an AI-guided approach to enhancing base editing efficiency by engineering Cas9 using ProMEP, a model previously developed by the authors. The authors optimized AncBE4max, a base editor, by introducing AI-predicted Cas9 mutations, ultimately developing AncBE4max-AI-8.3, which exhibited 2-3 times higher editing efficiency. This enhanced variant improved the performance of multiple base editors, including YEE-BE4max, CGBE, ABE-max, and ABE8e, in various cancer cell lines and human embryonic stem cells (hESCs). Mechanistic analysis, including catalytic activity tests and molecular dynamics (MD) simulations, revealed that AI-engineered Cas9 enhances both DNA binding affinity and catalytic activity without increasing off-target effects. This study highlights AI-driven protein engineering as a universal strategy for improving genome editing tools.

Overall, the work is well executed and nicely presented. I expect, with few minor improvements, this study would be sufficient for publication.

-Major comments-

1. Fitness distribution graphs of mutants would be very informative. Please provide a fitness histogram of single mutants, including the top 18 mutants. Additionally, a fitness histogram for triple mutants (including the top 10 mutants) would also offer deeper insights into the selection strategy.
2. The authors observed the catalytic activity of D10A Cas. However, H840A Cas is also widely used for broader applications. Providing data for H840A Cas would offer a more complete characterization of the final variant and be beneficial for the genome editing community.

-Minor comments-

1. The observed enrichment of X-K mutants appears redundant. Could this be due to an inherent structural or functional advantage of lysine substitutions, somehow? Providing a brief discussion on this trend would be helpful.
2. The authors describe that the difference in indel efficiency is "not significant" but do not provide p-values. Including statistical significance (p-values) would be necessary for a clear and rigorous presentation.

-Styles and typos-

1. Figure 2A: Line colors are not clearly distinguishable. Please use more distinct colors to improve clarity.
2. "After eliminating redundant mutation sites (e.g., for position 1280, where both G1218K and G1218R mutations were present, we retained the mutation with higher editing 3 efficiency), we selected G1218R, T622W, V1015K, C80K, and C574K to create a

new variant (Cas9-AI 5)(Figure 1A)." (position 1280 -> position 1218)

3. "Specifically, the enhancement effects of ABE-8e-AI-8.3 were mainly located at target sites with initial efficiencies below 80%, while sites with efficiencies above 85% reached an efficiency plateau for both ABE8e and ABE8e-AI 8.3 (Figure 3H, I)." (ABE-8e-AI-8.3 -> ABE8e-AI-8.3)

4. "Through NGS analysis, we compared the base editing levels at predicted off-target sites between the wild type and AI-AncBE4max groups (Figure EV11), which exhibited similar levels of off-target editing. Subsequently, we characterized the Cas9-independent off-target activity of the variant through orthogonal R-loop assays at six dead enAsCas12f R-loop sites." (Figure EV11 -> Figure EV12)

5. "Consistent with previous results, our high-performance variant showed similar off target activity compared with the AncBE4max editor (Figure EV12)." (Figure EV12 -> Figure EV13)

Dear Dr. Poonam Bheda,

Thank you for giving us the opportunity to revise our manuscript entitled “AI-guided Cas9 engineering provides an effective strategy to enhance base editing” (Manuscript Number: MSB-2025-12897-T) for publication in *Molecular Systems Biology*. We also thank the reviewers for their time, insightful comments, and constructive criticisms, which have significantly helped us improve the quality and clarity of our work.

We have carefully considered all the reviewers’ comments and revised the manuscript accordingly. Below is our detailed point-by-point response to each of the reviewers’ comments.

REVIEWER COMMENTS

Reviewer #1:

This study presents an AI-guided engineering approach to enhance base editing efficiency by optimizing the Cas9 protein. The researchers used the ProMEP tool to predict beneficial mutations in Cas9 and successfully developed a high-performance variant, AncBE4max-AI-8.3, which showed a 2-3-fold increase in average editing efficiency. This engineered Cas9 also effective in improving the performance of various base editors across multiple cell lines, demonstrated that AI models could serve as effective protein engineering tools for gene editing applications.

The study provides convincing evidence that the AI-engineered Cas9 variant can significantly enhance base editing efficiency. The significance of the progress of the study lies in the fact that the use of language models not only predicts the activity of single mutations, but also gives better prediction results in combination mutations.

However, the article also has the following problems that need to be explained or modified.

1. The article is written clearly and coherently as a whole, but the method part does not explain the principle and accuracy of the ProMEP algorithm, and it is necessary to fill in the top scores of single mutation and combined mutation prediction in the attached table. There are many AI tools with few and zero samples published, why this method is used, and whether it has advantages over other tools?

Response:

Thank you for your diligent review of our manuscript and the valuable advisement. We are sorry for missing a detailed introduction of ProMEP and its predicted fitness scores. In our prior study ^[1], we developed ProMEP, a multimodal method that enables zero-shot prediction of mutation effects. We selected this algorithm for Cas9 engineering based on three key advantages:

1) **ProMEP integrates both sequence and structure for mutation effects prediction.**

While protein language models (e.g., ESM1v ^[2], ProtGPT2 ^[3], Tranception ^[4]) emerge as a leading

solution for mutation effects prediction, they remain constrained by exclusive reliance on sequence data, overlooking structural information critical for protein function. In contrast, ProMEP employs a multimodal architecture to learn sequence-structure context from ~160 million proteins in the AlphaFold database. By integrating structural data containing spatial contact information, ProMEP enables multimodal mutation effect prediction.

2) ProMEP demonstrates state-of-the-art (SOTA) accuracy across diverse benchmarks

In our previous work ^[1], we have evaluated the performance of ProMEP on a series of benchmarks, Specifically, the ProteinGym benchmark covers 53 proteins and 1.43 million variants with experimentally determined function data. Leading mutation effect prediction methods were evaluated for comparison, including AlphaMissense ^[5] and ESM2-650M ^[6]. On the ProteinGYM benchmark, ProMEP achieves an average Spearman's rank correlation of 0.523, significantly outperforming the best protein language model, ESM2-650M (average Spearman's rank correlation of 0.465, P value = 0.03; t-test, two-sided). Besides, ProMEP demonstrates 2-3 orders of magnitude speed improvement with comparable performance than the best multimodal method, AlphaMissense (average Spearman's rank correlation of 0.523 vs 0.520, P value = 0.91; t-test, two-side). In the pathogenicity benchmark and stability benchmark, ProMEP consistently demonstrates SOTA performance.

3) ProMEP has successfully guided the engineering of multiple enzymes

We have successfully used ProMEP to guide the engineering of the gene-editing enzymes TnpB and TadA ^[1]. Specifically, a 5-site mutant of TnpB designed by ProMEP demonstrates a maximum gene-editing efficiency of 74.04% (vs 24.66% for the wild type). A 15-site mutant of TadA designed by ProMEP exhibits an A-to-G conversion frequency of up to 77.27% (vs 69.80% for ABE8e, a previous TadA-based adenine base editor) with significantly reduced bystander and off-target effects compared to ABE8e.

In summary, we chose ProMEP for Cas9 engineering based on its multimodal architecture, state-of-the-art performance, and demonstrated success in protein engineering applications. In addition, we have detailed the predicted probability distributions of all amino acid types at altered positions within Cas9 (Figure R1). ProMEP demonstrates high confidence in predicting top-ranked beneficial mutations, including C80K, C80R, V1015K, G1218R, and G1218K. We also provide the normalized fitness score predicted by ProMEP for your convenience (Table R1). Specifically, higher ProMEP scores (ranges from 0 to 1) indicate greater likelihood of beneficial mutation effects. In response to your comments, we have included these revisions in our manuscript (**Figure EV2 and Table EV1. Line 105-108, Line 368-378**). Thank you again for your valuable comments.

2. The experience of many past studies in the field of Cas9 activity optimization is that the positive charge mutation of R or K at the key position can improve the activity, but the prediction results of the software in Figure EV1 are concentrated on K, and the proportion of R is very small, and the experimental results also prove that the most important V1015 mutation is much higher than the activity of R than K, how to explain such a prediction bias?

Response:

Thank you very much for taking the time to thoroughly assess our manuscript. We hypothesize that the prediction bias originates from imbalanced amino acid distribution in natural Cas9 sequences. Using 123 Cas9 homologs (sequence identity <70%) from the Non-Redundant (NR) protein sequence database, we observed significantly higher lysine (K) prevalence than arginine (R) (10.94% vs 5.11%; Figure R3). As ProMEP was pre-trained on ~160 million proteins, it inherently learns evolutionary biases favoring naturally enriched residues.

During Cas9 engineering, ProMEP predicts amino acid distributions through sequence-structure context embedding. The observed natural K-enrichment in Cas9 homologs may predispose ProMEP to favor X-to-K mutations. Notably, ProMEP still identified diverse beneficial substitutions (e.g., G1218R, T622W, V1015K, C80M), demonstrating its capacity to transcend evolutionary biases while leveraging conserved fitness patterns.

We discuss this evolutionary prediction bias in the revised manuscript (Lines 325-334).

3. The study mainly focuses on the improvement of editing efficiency, and the experimental results showed that the off-target effect was similar to that of the wild type. However, the clinical application of wild-type SpCas9 is limited due to its high off-target rate, and SpCas9-HF, a low-off-target mutant, has long been studied. If the four REC domain mutations of SpCas9-HF1 are introduced in experimental, it is of practical significance to optimize the variants with lower off-target rate.

Response:

Thank you for your valuable comments. Following your suggestions, we introduced the four REC domain mutations of SpCas9-HF1 into AncBE4max and AI-AncBE4max, and evaluated their editing efficiency and off-target effects (Figure R4). The results showed that the combination of HF1 mutations with our eight-mutation set indeed enabled the BE to achieve both high editing efficiency and excellent low off-target performance. As you mentioned, the introduction of high-fidelity mutations will significantly improve the practical applicability of our BE variants. The relevant content has been added to the revised version (Figure 3J, K, EV3. Lines 215-228).

4. A minor points in part of result, AI-AncBE4max groups (Figure EV11) need to be changed to EV12.

Response:

We thank the reviewer for their careful reading. We confirm the presence of this labeling inconsistency and have implemented the correction in the revised manuscript.

Reviewer #2:

Wei et al. present an AI-guided approach to enhancing base editing efficiency by engineering Cas9 using

ProMEP, a model previously developed by the authors. The authors optimized AncBE4max, a base editor, by introducing AI-predicted Cas9 mutations, ultimately developing AncBE4max-AI-8.3, which exhibited 2-3 times higher editing efficiency. This enhanced variant improved the performance of multiple base editors, including YEE-BE4max, CGBE, ABE-max, and ABE8e, in various cancer cell lines and human embryonic stem cells (hESCs). Mechanistic analysis, including catalytic activity tests and molecular dynamics (MD) simulations, revealed that AI-engineered Cas9 enhances both DNA binding affinity and catalytic activity without increasing off-target effects. This study highlights AI-driven protein engineering as a universal strategy for improving genome editing tools.

Overall, the work is well executed and nicely presented. I expect, with few minor improvements, this study would be sufficient for publication.

-Major comments-

1. Fitness distribution graphs of mutants would be very informative. Please provide a fitness histogram of single mutants, including the top 18 mutants. Additionally, a fitness histogram for triple mutants (including the top 10 mutants) would also offer deeper insights into the selection strategy.

Response:

Thank you for your insightful comments. We have added fitness landscape for all single mutants and top 10,000 triple mutants (Figure R2). The y-axis corresponding to the normalized fitness score predicted by ProMEP. Specifically, higher ProMEP scores (ranges from 0 to 1) indicate greater likelihood of beneficial mutation effects. During Cas9 engineering, we first calculated the normalized fitness score of all single mutants and ranked them accordingly. We then analyzed the mutation-type enrichment in top 5% of these ranked mutants. To transcend evolutionary biases while leveraging conserved fitness patterns, we selected 18 high-fitness single mutants based on the enrichment analysis, comprising:

- The top 6 X-to-K mutants
- The top 3 X-to-C mutants
- The top 3 X-to-W mutants
- The top 3 X-to-M mutants
- The top 3 X-to-R mutants

For triple mutants, we selected the top 10 candidates based on normalized fitness scores. We also provide the normalized fitness score of top 18 single mutants and top 10 triple mutants for your convenience (Table R1). In addition, we detailed the predicted probability distributions of all amino acid types at altered positions within Cas9 (Figure R1). ProMEP demonstrates high confidence in predicting top-ranked beneficial mutations, including C80K, C80R, V1015K, G1218R, and G1218K. We appreciate your valuable comments and include a detailed description of the selection strategy in our revised manuscript (Figure EV1A, EV2, Table EV1. Line 325-334, Line 386-391).

2. The authors observed the catalytic activity of D10A Cas. However, H840A Cas is also widely used for broader applications. Providing data for H840A Cas would offer a more complete characterization of the final variant and be beneficial for the genome editing community.

Response:

Thank you for your suggestions. Adding data on the H840A Cas variant indeed helps provide a more comprehensive description of our variants. In this revision, we constructed H840A Cas9 and AI-H840A Cas9 and observed an enhancement in cleavage activity similar to that of AI-D10A Cas9. This further supports that our engineering strategy successfully enhances Cas9 activity. The corresponding results have been included in the revised manuscript (Appendix Figure S11).

-Minor comments-

1. The observed enrichment of X-K mutants appears redundant. Could this be due to an inherent structural or functional advantage of lysine substitutions, somehow? Providing a brief discussion on this trend would be helpful.

Response:

Thank you for your diligent review of our manuscript. We attribute the observed X-to-K mutant enrichment to prediction biases arising from imbalanced amino acid distributions in natural Cas9 sequences. Using 123 Cas9 homologs (sequence identity <70%, Figure R1) from the Non-Redundant (NR) protein sequence database, we found lysine (K) accounts for 10.94% of residues (vs arginine [R]=5.11%, histidine [H]=2.01%, aspartate [D]=6.51%).

As ProMEP was pre-trained on ~160 million proteins, it internalizes evolutionary biases favoring naturally abundant residues. When predicting fitness scores for Cas9 mutants through sequence-structure context embedding, the observed K-enrichment in Cas9 homologs may predispose ProMEP to favor X-to-K mutations. Notably, ProMEP still identified diverse beneficial substitutions (e.g., G1218R, T622W, V1015K, C80M), demonstrating its capacity to transcend evolutionary biases while leveraging conserved fitness patterns.

We have now added a discussion in our revised manuscript (Line 326-334).

2. The authors describe that the difference in indel efficiency is "not significant" but do not provide p-values. Including statistical significance (p-values) would be necessary for a clear and rigorous presentation.

Response:

Thank you for your valuable comments. For the data regarding the indel efficiency described as "not significant" in the manuscript, we have added the corresponding p-values in the figure legend.

-Styles and typos-

1. Figure 2A: Line colors are not clearly distinguishable. Please use more distinct colors to improve clarity.

2. "After eliminating redundant mutation sites (e.g., for position 1280, where both G1218K and G1218R mutations were present, we retained the mutation with higher editing efficiency), we selected G1218R,

T622W, V1015K, C80K, and C574K to create a new variant (Cas9-AI 5) (Figure 1A)." (position 1280 -> position 1218)

3. "Specifically, the enhancement effects of ABE-8e-AI-8.3 were mainly located at target sites with initial efficiencies below 80%, while sites with efficiencies above 85% reached an efficiency plateau for both ABE8e and ABE8e-AI 8.3 (Figure 3H, I)." (ABE-8e-AI-8.3 -> ABE8e-AI-8.3)

4. "Through NGS analysis, we compared the base editing levels at predicted off-target sites between the wild type and AI-AncBE4max groups (Figure EV11), which exhibited similar levels of off-target editing. Subsequently, we characterized the Cas9-independent off-target activity of the variant through orthogonal R-loop assays at six dead enAsCas12f R-loop sites." (Figure EV11 -> Figure EV12)

5. "Consistent with previous results, our high-performance variant showed similar off target activity compared with the AncBE4max editor (Figure EV12)." (Figure EV12 -> Figure EV13)

Response:

We thank the reviewer for their rigorous attention to manuscript details. We have completed the following comprehensive revisions: (1) The color scheme in Figure 2A has been optimized using higher-contrast colors to ensure clear visual discrimination between data groups; (2) All documentation and labeling errors noted in points #2-5 (specifically position 1280 → 1218 and ABE-8e-AI-8.3 → ABE8e-AI-8.3) have been corrected and we updated the figure designations in the revised manuscript and conducted a thorough recheck of the writing and numbering throughout the document to ensure accuracy.

1. Cheng, P., et al., *Zero-shot prediction of mutation effects with multimodal deep representation learning guides protein engineering*. Cell Res, 2024. **34**(9): p. 630–647.
2. Meier, J., et al., *Language models enable zero-shot prediction of the effects of mutations on protein function*. Advances in neural information processing systems, 2021. **34**: p. 29287–29303.
3. Ferruz, N., S. Schmidt, and B. Hocker, *ProtGPT2 is a deep unsupervised language model for protein design*. Nat Commun, 2022. **13**(1): p. 4348.
4. Notin, P., et al. *Tranception: protein fitness prediction with autoregressive transformers and inference-time retrieval*. in *International Conference on Machine Learning*. 2022. PMLR.
5. Cheng, J., et al., *Accurate proteome-wide missense variant effect prediction with AlphaMissense*. Science, 2023. **381**(6664): p. eadg7492.
6. Lin, Z., et al., *Evolutionary-scale prediction of atomic-level protein structure with a language model*. Science, 2023. **379**(6637): p. 1123–1130.

Response figures

Figure R1 The predicted probability distributions of all amino acid types at altered positions within Cas9.

A

Figure R2 Normalized fitness score predicted by ProMEP.

Figure R3 Sequence logo plot of Cas9 homologs.

Figure R4 Editing efficiency and off-target effects of HF1-BEs.

Figure R5 Cleavage activity of Cas9(H840A)

Table R1. Normalized fitness score of single mutations and combined mutations predicted by ProMEP.

Mutation	Normalized fitness score
C80K	0.832
C80R	0.818
G1218R	0.771
G1218K	0.765
V1015K	0.754
M1169K	0.751
C574K	0.748
P1090K	0.736
M1169R	0.724
H723M	0.702
C80M	0.701
N77W	0.693
C80W	0.677
T622W	0.652

C574M	0.642
R100C	0.575
A262C	0.574
D821C	0.567
AncBE4Max-AI-8.1 (C80K:G1218R:T622W:C574K:V1015K:N501Y:M939F:H723V)	1.000
AncBE4Max-AI-8.2 (C80K:G1218R:T622W:C574K:V1015K:N501Y:M939F:M1169K)	0.974
AncBE4Max-AI-8.3 (C80K:G1218R:T622W:C574K:V1015K:N501Y:M939F:H723I)	0.957
AncBE4Max-AI-8.4 (C80K:G1218R:T622W:C574K:V1015K:N501Y:M939F:H723L)	0.953
AncBE4Max-AI-8.5 (C80K:G1218R:T622W:C574K:V1015K:N501Y:M939F:V824T)	0.935
AncBE4Max-AI-8.6 (C80K:G1218R:T622W:C574K:V1015K:N501Y:M939F:P1090K)	0.929
AncBE4Max-AI-8.7 (C80K:G1218R:T622W:C574K:V1015K:N501Y:M939F:P1137T)	0.922
AncBE4Max-AI-8.8 (C80K:G1218R:T622W:C574K:V1015K:N501Y:M939F:N980D)	0.915
AncBE4Max-AI-8.9 (C80K:G1218R:T622W:C574K:V1015K:N501Y:M939F:G485E)	0.907
AncBE4Max-AI-8.10 (C80K:G1218R:T622W:C574K:V1015K:N501Y:M939F:P28I)	0.906

8th Aug 2025

Manuscript Number: MSB-2025-12897R

Title: AI-guided Cas9 engineering provides an effective strategy to enhance base editing

Dear Dr. Wei,

Thank you for the submission of your revised manuscript to Molecular Systems Biology. I am pleased to inform you that we will be able to accept your manuscript pending the following final amendments and appropriate response to reviewers:

1) Please note that all corresponding authors are required to supply an ORCID ID for their name. Currently Dr. Yu Zhang does not have an ORCID ID associated with their profile. An email requesting the ORCID ID to be linked with instructions was sent to Dr. Zhang on July 10, 2025.

2) Please update the README file on your ProMEP Github page with practical use instructions for potential future users of your code/model.

3) In the main manuscript file, please format the Data availability section describing how the data, code etc. have been made available according to the example below:

"The datasets and computer code produced in this study are available in the following databases:

- Chip-Seq data: Gene Expression Omnibus GSE46748 (<https://www.ncbi.nlm.nih.gov/geo/query/acc.cgi?acc=GSE46748>)

- Modeling computer scripts: GitHub (<https://github.com/SysBioChalmers/GECKO/releases/tag/v1.0>)

- [data type]: [full name of the resource] [accession number/identifier] ([doi or URL or identifiers.org/DATABASE:ACCESSION])"

4) Our journal encourages inclusion of *data citations in the reference list* to directly cite datasets that were re-used and obtained from public databases. Data citations in the article text are distinct from normal bibliographical citations and should directly link to the database records from which the data can be accessed. In the main text, data citations are formatted as follows: "Data ref: Smith et al, 2001" or "Data ref: NCBI Sequence Read Archive PRJNA342805, 2017". In the Reference list, data citations must be labeled with "[DATASET]". A data reference must provide the database name, accession number/identifiers and a resolvable link to the landing page from which the data can be accessed at the end of the reference. Further instructions are available at .

5) In the Methods, please take care of the following:

- Please be sure to include a sentence in the Methods as to whether or not the cell lines were recently authenticated and tested for mycoplasma contamination. Please also be sure to update the Author Checklist with this information and where it can be found in the manuscript.

- Please ensure that a statement on whether or not blinding was done is included in the Methods even if no blinding was done. Please also be sure to update the Author Checklist with this information and where it can be found in the manuscript.

7) Please place individual sections of the manuscript in the following order: Title page - Abstract & Keywords - Introduction - Results - Discussion - Methods - Data Availability - Acknowledgements - Disclosure and Competing Interests Statement - References - Figure Legends - Expanded View Figure Legends.

8) For the figures and figure legends, please take care of the following:

- Each main and EV figure should fit onto a single page. Currently most of the figures are too large. The figures do not need to be zipped together, but rather should be uploaded as individual files.

- Please make sure to update the callouts of all figures in the main manuscript text. Currently a callout is missing for Appendix Figure S10.

- Please note that information related to n is missing in the legends of figure 4E.

9) Please remove the legend for Table EV1 from the manuscript. Rather the legend should appear in the same sheet/tab in the Excel file (currently this is provided in a separate sheet/tab).

10) In the Appendix file, please remove the author list and affiliations, as this is not needed.

11) Please ensure that all funding sources are entered into the manuscript submission system. Currently local grant [CXPJH122006-1014] appears to be missing.

12) Please shorten the synopsis text standfirst to a maximum of 300 characters, including spaces.

13) Source Data should not be uploaded in a single zipped folder, but rather should be organized as a single source data file (zipped) per figure for main figures (all EV and/or Appendix figure Source Data can be included in a single folder), with the panels clearly visible in the folder structure instead of a single excel file for all Source Data. e.g. all the Source data files for figure 1 need to be saved in a single folder and this needs to be zipped and then uploaded as "SD figure 1.zip" file.

14) As part of the EMBO Publications transparent editorial process initiative (see our policy here:

https://www.embopress.org/transparent-process#Review_Process), Molecular Systems Biology will publish online a Peer Review File (PRF) to accompany accepted manuscripts. This file will be published in conjunction with your paper and will include the anonymous referee reports, your point-by-point response and all pertinent correspondence relating to the manuscript. Let us know whether you agree with the publication of the PRF and as here, if you want to remove or not any figures from it prior to publication. Please note that the Authors checklist will be published at the end of the PRF.

15) After your paper is published, we may promote it on social media. If you have any handles or hashtags for Bluesky you would like included, please let us know.

16) Please provide a point-by-point letter INCLUDING my comments as well as the reviewer's reports and your detailed responses (as Word file).

I look forward to reading a new revised version of your manuscript as soon as possible.

Yours sincerely,

Poonam Bheda, PhD
Scientific Editor
Molecular Systems Biology

Reviewer #1:

The authors' responses and manuscript revisions have adequately addressed the reviewers' concerns. The current data robustly support the conclusions, and the work meets publication standards with compelling innovation and translational value. However, minor refinements are warranted before final acceptance:

1. While ProMEP demonstrates competitive performance on ProteinGYM (Spearman's $\rho=0.523$, significantly outperforming ESM2-650M and closely rivaling top tools), additional validation specific to Cas9 point mutations is essential. We request quantitative comparison between ProMEP and other leading tools using both the ProteinGYM Cas9 dataset and the authors' experimental data for the 18 validated single-point mutants.
2. The significance of enhanced HF-BE efficiency must be explicitly framed in the Introduction and Discussion. Introduction: Highlight the long-standing efficacy-safety trade-off in high-fidelity editors. Discussion: Emphasize that overcoming this trade-off via AI engineering represents a critical advance for safe in vivo applications. Cite clinical relevance and declare this as the first AI-optimized editor achieving dual high efficiency and low off-target effects.
3. Explicitly state whether the 18 high-ranking mutations were selected based on top fitness scores or via a hybrid approach combining score thresholds with mutation-type quotas (as implied in Methods).

Reviewer #2:

The authors have provided a thoughtful response to all my prior comments and have improved the manuscript. I appreciate the addition of fitness distribution data for both single and triple mutants, as well as the inclusion of functional validation data for the H840A Cas9 variant. These additions strengthen the mechanistic depth and broaden the applicability of the study.

Overall, this revised version convincingly demonstrates the potential of AI-guided Cas9 engineering as a broadly applicable strategy to enhance base editing. The integration of mechanistic insights, broad validation across base editor classes, and attention to off-target effects support the robustness and relevance of the findings.

I am satisfied with the revisions and now support the publication of this manuscript in Molecular Systems Biology.

Title: AI-guided Cas9 engineering provides an effective strategy to enhance base editing

Author: Dongyi Wei, et al.

Number: MSB-2025-12897RR

Dear Editor and Reviewers,

We sincerely thank the editor and reviewers for their valuable comments. We have carefully supplemented and revised the manuscript as requested. Below is our detailed point-by-point response.

EDITOR COMMENTS

1) Please note that all corresponding authors are required to supply an ORCID ID for their name. Currently Dr. Yu Zhang does not have an ORCID ID associated with their profile. An email requesting the ORCID ID to be linked with instructions was sent to Dr. Zhang on July 10, 2025.

Dr. Yu Zhang has already replied to the relevant email and provided her ORCID.

2) Please update the README file on your ProMEP Github page with practical use instructions for potential future users of your code/model.

The README file has been updated accordingly.

We've released code for ProMEP-guided protein engineering featuring:

- Virtual single-point saturation mutagenesis library generation
- Fitness score calculation for all mutants
- Mutant ranking

3) In the main manuscript file, please format the Data availability section describing how the data, code etc. have been made available according to the example below:

"The datasets and computer code produced in this study are available in the following databases:

- Chip-Seq data: Gene Expression Omnibus GSE46748

(<https://www.ncbi.nlm.nih.gov/geo/query/acc.cgi?acc=GSE46748>)

- Modeling computer scripts: GitHub

(<https://github.com/SysBioChalmers/GECKO/releases/tag/v1.0>)

- [data type]: [full name of the resource] [accession number/identifier] ([doi or URL or identifiers.org/DATABASE:ACCESSION])"

We have revised our Data Availability section according to the example provided.

4) Our journal encourages inclusion of *data citations in the reference list* to directly cite datasets that were re-used and obtained from public databases. Data citations in the article text are distinct from normal bibliographical citations and should directly link to the database records from which the data can be accessed. In the main text, data citations are formatted as follows: "Data ref: Smith et al, 2001" or "Data ref: NCBI Sequence Read Archive PRJNA342805, 2017". In the Reference list, data citations must be labeled with "[DATASET]". A data reference must provide the database name, accession number/identifiers and a resolvable link to the landing page from which the data can be accessed at the end of the reference. Further instructions are available at <https://www.embopress.org/page/journal/17574684/authorguide#referencesformat>.

We have updated the reference list and added data citation following the guidelines.

5) In the Methods, please take care of the following:

- Please be sure to include a sentence in the Methods as to whether or not the cell lines were recently authenticated and tested for mycoplasma contamination. Please also be sure to update the Author Checklist with this information and where it can be found in the manuscript.
- Please ensure that a statement on whether or not blinding was done is included in the Methods even if no blinding was done. Please also be sure to update the Author Checklist with this information and where it can be found in the manuscript.

We have added and highlighted these two statements, and also updated the relevant content in the Author Checklist.

We have removed the Reagents and Tools Table from the manuscript and submitted it as a separated file.

7) Please place individual sections of the manuscript in the following order: Title page - Abstract & Keywords - Introduction - Results - Discussion - Methods - Data Availability - Acknowledgements - Disclosure and Competing Interests Statement - References - Figure Legends - Expanded View Figure Legends.

We have adjusted the order of sections according to the order you commented.

8) For the figures and figure legends, please take care of the following:

- Each main and EV figure should fit onto a single page. Currently most of the figures are too large. The figures do not need to be zipped together, but rather should be uploaded as

individual files.

- Please make sure to update the callouts of all figures in the main manuscript text. Currently a callout is missing for Appendix Figure S10.

- Please note that information related to n is missing in the legends of figure 4E.

We have added a callout for Appendix Figure S10 in the main manuscript and noted the n information in the legend of Figure 4E. Additionally, we have uploaded each main figure and EV figure as separate files.

9) Please remove the legend for Table EV1 from the manuscript. Rather the legend should appear in the same sheet/tab in the Excel file (currently this is provided in a separate sheet/tab).

We have removed the legend for Table EV1 from the manuscript and included it in the same sheet in the Excel file.

10) In the Appendix file, please remove the author list and affiliations, as this is not needed.

We removed the author list and affiliations in the Appendix file.

11) Please ensure that all funding sources are entered into the manuscript submission system. Currently local grant [CXPJJH122006-1014] appears to be missing.

We have entered this local grant into the submission system.

12) Please shorten the synopsis text standfirst to a maximum of 300 characters, including spaces.

We have revised the synopsis text and shortened it less than 300 characters.

13) Source Data should not be uploaded in a single zipped folder, but rather should be organized as a single source data file (zipped) per figure for main figures (all EV and/or Appendix figure Source Data can be included in a single folder), with the panels clearly visible in the folder structure instead of a single excel file for all Source Data. e.g. all the Source data files for figure 1 need to be saved in a single folder and this needs to be zipped and then uploaded as "SD figure 1.zip" file.

We have reorganized the source data folders and uploaded them individually as required.

14) As part of the EMBO Publications transparent editorial process initiative (see our policy here: https://www.embopress.org/transparent-process#Review_Process), Molecular Systems Biology will publish online a Peer Review File (PRF) to accompany accepted manuscripts. This file will be published in conjunction with your paper and will include the anonymous referee reports, your point-by-point response and all pertinent correspondence relating to the manuscript. Let us know whether you agree with the publication of the PRF and as here, if

you want to remove or not any figures from it prior to publication. Please note that the Authors checklist will be published at the end of the PRF.

We agree with the publication of the PRF. No changes are needed.

15) After your paper is published, we may promote it on social media. If you have any handles or hashtags for Bluesky you would like included, please let us know.

16) Please provide a point-by-point letter INCLUDING my comments as well as the reviewer's reports and your detailed responses (as Word file).

REVIEWER COMMENTS

Reviewer #1

The authors' responses and manuscript revisions have adequately addressed the reviewers' concerns. The current data robustly support the conclusions, and the work meets publication standards with compelling innovation and translational value. However, minor refinements are warranted before final acceptance:

1. While ProMEP demonstrates competitive performance on ProteinGYM (Spearman's $\rho=0.523$, significantly outperforming ESM2-650M and closely rivaling top tools), additional validation specific to Cas9 point mutations is essential. We request quantitative comparison between ProMEP and other leading tools using both the ProteinGYM Cas9 dataset and the authors' experimental data for the 18 validated single-point mutants.

Thank you for your diligent review and valuable feedback on our manuscript. We evaluated ProMEP's mutation effect prediction performance on two datasets: the ProteinGym Cas9 benchmark (*Cas9_ProteinGym*, containing 8,117 single-point mutants) and an experimental dataset (*Cas9_18mutants*, containing 18 single-point mutants). Notably, five mutants (*C80R*, *C80W*, *G1218R*, *M1169K* and *M1169R*) overlap between datasets. Current leading methods, including ESM2-650M, ESM2-3B, Tranception (Large, no retrieval)^[1] and ProstT5^[2], are used for comparison. While all methods showed comparable performance on *Cas9_ProteinGym*, ProMEP achieves a Spearman's ρ of 0.736 on *Cas9_18mutants*, outperforming ESM2-650M ($\rho=0.404$) and Tranception ($\rho=0.251$). We attribute ProMEP's performance on *Cas9_18mutants* to the hybrid strategy combining fitness score thresholds with mutation-type quotas. The small sample size of *Cas9_18mutants* may also contribute to performance variability. We appreciate your valuable comments and have incorporated quantitative comparisons in our revised manuscript (Line 342-346, 404-414).

Zero-shot prediction of mutation effects on Cas9 datasets.

2. The significance of enhanced HF-BE efficiency must be explicitly framed in the Introduction and Discussion. Introduction: Highlight the long-standing efficacy-safety trade-off in high-fidelity editors. Discussion: Emphasize that overcoming this trade-off via AI engineering represents a critical advance for safe in vivo applications. Cite clinical relevance and declare this as the first AI-optimized editor achieving dual high efficiency and low off-target effects.

Thank you for your thoughtful feedback. We appreciate your suggestion to explicitly highlight the significance of enhanced HF-BE efficiency and the long-standing efficacy-safety trade-off in high-fidelity editors. In response, we have revised the Introduction to emphasize the challenges associated with this trade-off and its critical implications for therapeutic genome editing. Additionally, we have updated the Discussion to underscore how our AI-engineered approach successfully addresses this trade-off, representing a significant advancement for safe in vivo applications. These revisions aim to better frame the importance of our findings in the context of genome editing advancements. Thank you again for your valuable input, which has greatly improved the clarity and impact of our manuscript. The relevant content has already been presented in the revised manuscript (Line 80-85, 352-257)

3. Explicitly state whether the 18 high-ranking mutations were selected based on top fitness scores or via a hybrid approach combining score thresholds with mutation-type quotas (as implied in Methods).

Thank you for your valuable comments. Following your suggestions, we have stated that 18 high-ranking mutations were selected via a hybrid approach combining fitness score thresholds with mutation-type quotas. The relevant content has been added to the revised version (Lines 126-127, 342).

Reviewer #2

The authors have provided a thoughtful response to all my prior comments and have improved the manuscript. I appreciate the addition of fitness distribution data for both single and triple mutants, as well as the inclusion of functional validation data for the H840A Cas9 variant. These additions strengthen the mechanistic depth and broaden the applicability of the study.

Overall, this revised version convincingly demonstrates the potential of AI-guided Cas9 engineering as a broadly applicable strategy to enhance base editing. The integration of mechanistic insights, broad validation across base editor classes, and attention to off-target effects support the robustness and relevance of the findings.

I am satisfied with the revisions and now support the publication of this manuscript in *Molecular Systems Biology*.

Thank you for your thoughtful and positive feedback on our revised manuscript. Your comments and suggestions have significantly improved the clarity and robustness of our work, and we are grateful for your constructive input throughout the review process. Thank you again for your time and effort in reviewing our manuscript.

[1] Notin P, Dias M, Frazer J, Marchena-Hurtado J, Gomez AN, Marks D, Gal Y, 2022. *Tranception: protein fitness prediction with autoregressive transformers and inference-time retrieval*, *International Conference on Machine Learning*. PMLR, pp. 16990–17017.

[2] Heinzinger M, Weissenow K, Sanchez Joaquin G, Henkel A, Mirdita M, Steinegger M, Rost B (2024) *Bilingual language model for protein sequence and structure*. *NAR Genomics and Bioinformatics* 6

21st Aug 2025

Manuscript number: MSB-2025-12897RR

Title: AI-guided Cas9 engineering provides an effective strategy to enhance base editing

Dear Dr. Wei,

Thank you again for sending us your revised manuscript. We are now satisfied with the modifications made and I am pleased to inform you that your paper has been accepted for publication.

Yours sincerely,

Poonam Bheda, PhD
Scientific Editor
Molecular Systems Biology
